Manuscript prepared for Atmos. Chem. Phys.
with version 2014/09/16 7.15 Copernicus papers of the LATEX class copernicus.cls.
Date: 8 June 2020

# Inconsistencies between chemistry climate model and observed lower stratospheric ozone trends since 1998

William T. Ball[1,2,3], Gabriel Chiodo[2,4], Marta Abalos[5], Justin Alsing[6,7], and
Andrea Stenke[2]

[1]Department of Geoscience and Remote Sensing, Faculty of Civil Engineering and Geosciences,
TU Delft, Stevinweg 1, 2628 CN Delft, The Netherlands

[2]Institute for Atmospheric and Climate Science, Swiss Federal Institute of Technology Zurich,
Universitaetstrasse 16, CHN, CH-8092 Zurich, Switzerland

[3]Physikalisch-Meteorologisches Observatorium Davos World Radiation Centre, Dorfstrasse 33,
7260 Davos Dorf, Switzerland

[4]Department of Applied Physics and Applied Mathematics, 5 Columbia University, New York, NY,
USA

[5]Earth Physics and Astrophysics Dep., Universidad Complutense de Madrid, Avda. Complutense
s/n, 28040 Madrid, Spain

[6]Oskar Klein Centre for Cosmoparticle Physics, Stockholm University, Stockholm SE-106 91,
Sweden

[7]Imperial Centre for Inference and Cosmology, Department of Physics, Imperial College London,
Blackett Laboratory, Prince Consort Road, London SW7 2AZ, UK

*Correspondence to:* W. T. Ball (w.t.ball@tudelft.nl)

**Abstract.** The stratospheric ozone layer shields surface life from harmful ultraviolet radiation. Following the Montreal Protocol ban of long-lived ozone depleting substances (ODSs), rapid depletion of total column ozone (TCO) ceased in the late 1990s and ozone above 32 km now enjoys a clear recovery. However, there is still no confirmation of TCO recovery, and evidence has emerged that ongoing quasi-global (60°S–60°N) lower stratospheric ozone decreases may be responsible, dominated by low latitudes (30°S–30°N). Chemistry climate models (CCMs) used to project future changes predict that lower stratospheric ozone will decrease in the tropics by 2100, but not at mid-latitudes (30°–60°). Here, we show that CCMs display an ozone decline similar to that observed in the tropics over 1998–2016, likely driven by ~~a~~ **an** increase of tropical upwelling. On the other hand, mid-latitude lower stratospheric ozone is observed to decrease, while CCMs **that specify real-world historical meteorological fields show instead an increase up to present day. However, these cannot be used to simulate future changes; we demonstrate here that free-running CCMs used for projections also show increases**. Despite opposing lower stratospheric ozone changes, which should induce opposite temperature trends, CCM and observed temperature trends agree; we demonstrate that opposing model-observation stratospheric water vapour (SWV) trends, and their associated radiative effects, explain why temperature changes agree in spite of opposing ozone trends. We provide new evidence that the observed mid-latitude trends can be explained by enhanced mixing between the tropics and extratropics. We further show that the temperature trends are consistent with the observed mid-latitude ozone decrease. Together, our results suggest that large scale circulation changes expected in the future from increased greenhouse gases (GHGs) may now already be underway, but that most CCMs are not simulating well mid-latitude ozone layer changes. **However, it is important to emphasize that the periods considered here are short and internal variability that is both intrinsic to each CCM and different to observed historical variability is not well characterised and can influence trend estimates. Nevertheless,** the reason CCMs do not exhibit the observed changes ~~urgently~~ needs to be ~~understood to improve~~ *identified to allow models to be improved in order to build* confidence in future projections of the ozone layer.

## 1 Introduction

In the latter half of the 20[th] Century, emissions of halogen-containing ozone depleting substances (ODSs) led to a decline of the ozone layer at all latitudes across the globe (WMO, 2014). Following the almost universal implementation of the Montreal Protocol and its amendments (MPA) by governments, production of ODSs halted soon after and ODS loading in the atmosphere peaked in the mid-to-late 1990s (Newman et al., 2007; Chipperfield et al., 2017). By 1998, quasi-global (60°S–60°N) total column ozone had globally declined by ~5%, and spring-time ozone over the Antarctic regularly saw losses of two-thirds in the total column (WMO, 2018). In subsequent years, it emerged that global total column ozone levels had stopped falling by around 1998–2000 thanks to the MPA

(WMO, 2006), and research has turned to identifying an ozone recovery related to ODS declines (Chipperfield et al., 2017). In the upper stratosphere (1–10 hPa; 32–48 km), ozone now enjoys a clear recovery with levels now significantly above those of 1998 (Bourassa et al., 2017; Sofieva et al., 2017; Steinbrecht et al., 2017; Ball et al., 2017; WMO, 2018; Petropavlovskikh et al., 2019).

The area of the Antarctic ozone hole during September and October is now also showing signs of year-on-year shrinkage (Solomon et al., 2016; Pazmino et al., 2017; WMO, 2018). As such, there are clear indications that the MPA has worked in reducing atmospheric ODSs, that further significant and serious depletion of the ozone layer has been avoided (Egorova et al., 2013; Chipperfield et al., 2015), and that some regions exhibit an MPA-dependent recovery.

However, the picture has become more complicated, particularly in the lower stratosphere. Recent findings indicate that contrary to **chemistry climate models (CCMs) using historical meteorology to account for dynamical variability, and the multi-model mean (MMM) from CCM projections** ~~chemistry climate model (CCM) predictions~~, ozone in the lower stratosphere **does not yet display ozone increases since the turn of the century Steinbrecht et al. (2017); Petropavlovskikh**

**et al. (2019), and indeed there is evidence that it may have** ~~has~~ continued to decrease over 1998-2016 (Ball et al., 2018; Zerefos et al., 2018; Wargan et al., 2018; Ball et al., 2019; Orbe et al., 2020) and is offsetting the increases in the upper stratosphere (Ball et al., 2018). Changes and variability related to dynamics have been proposed as a mechanism, with evidence from reanalysis data (Wargan et al., 2018; Orbe et al., 2020) and a chemistry transport model (CTM) (Chipperfield et al., 2018).

Rising tropospheric ozone (Ziemke et al., 2018; Gaudel et al., 2018) also interferes in clearly detecting an ozone layer increase when considering total column ozone alone as a proxy for the ozone layer (Ball et al., 2018). A statistically significant increase in total column ozone since 1998 or 2000 remains undetected (Weber et al., 2018).

A confounding factor in detecting an ODS-related recovery is that rising greenhouse gas (GHG)

concentrations also affect the apparent recovery rate in the stratosphere through two main processes. Increased GHGs lead to a cooling of the stratosphere, thereby slowing temperature-dependent catalytic reaction rates that destroy ozone, and ~50% of the upper stratospheric ozone increase has been attributed to GHG-induced temperature decreases (WMO, 2014). Rising GHGs are also expected to modify the wave-driving of the large-scale Brewer-Dobson circulation (BDC) (see Butchart (2014)

and references therein), mainly through an acceleration of tropical upwelling that CCMs robustly simulate in projections towards the end of the 21st Century. This upwelling is correlated with a decline in tropical lower stratospheric ozone (SPARC/WMO, 2010) that means, by 2100, total column ozone in the tropics will not have recovered to pre-1980s levels (Eyring et al., 2010; Dhomse et al., 2018). However, it has not been demonstrated, using CCMs, that mid-latitude (30°–60°) lower strato-

spheric ozone should decrease and, indeed, MMM estimates that aggregate multiple CCMs indicate positive, though non-significant, changes at mid-latitudes by 2013 (WMO, 2014) or 2016 (WMO, 2018).

It has been proposed that the decline in ozone detected at mid-latitudes is a consequence of large natural variability interfering with linear regression trend analysis (Stone et al., 2018; Chipperfield et al., 2018). A southern hemisphere (SH) increase in ozone in 2017 was simulated using a chemistry transport model (CTM) to exceed the estimated long-term decrease over the previous 19 years when integrated over the quasi-global lower stratosphere (Chipperfield et al., 2018). When observations ~~became available~~ **were analysed**, this short-term large increase in ozone was found to be ~60% of the modelled change (Ball et al., 2019), and 2018 saw quasi-global ozone begin to decrease again towards the end of the year; a seasonal dependence of the quasi-biennial oscillation (QBO) has been implicated as the primary driver of these mid-latitude changes, i.e. dynamically driven. Due to the absence of an interaction of seasonal and QBO terms in the regression analysis, such non-linearities are not considered and the large dynamical changes are not accounted for, leading to large residuals that can indeed influence trend terms. In the particular case of 2017, while the magnitude and probability of the inferred negative ozone change for 1998–2017 in the SH lower stratosphere has reduced relative to 1998–2016, it remains negative; equatorial and NH changes remain negative with similar confidence over the last few years.

The aforementioned CTM (Chipperfield et al., 2018) drives the dynamics, temperature and surface level pressure using reanalysis (Dee et al., 2011) – a coherent, historical assimilation of observations using a general circulation model – that aims to reproduce historical behaviour of the atmosphere as closely as possible to compare with observations. The chemistry, however, is allowed to evolve freely and, generally, a CTM can simulate the observed behaviour of ozone reasonably well. It has also been shown that two state-of-the-art CCMs that use reanalyses in specified-dynamics (SD) mode – that is to guide, but not govern, the dynamics of models – do not reproduce the changes seen in the lower stratosphere (Ball et al., 2018). This is despite the aim of such models to reproduce historical dynamical changes while allowing freedom for the models to evolve in their own, model-dependent way. Why they do not reproduce the observations remains an open question. At the other end of the model spectrum are free-running (FR) CCMs – with no interference from reanalyses in governing dynamics. FR CCMs are used to investigate how the atmosphere responds to different forcing scenarios, and future projections of GHG and ODS changes (WMO, 2018); in this mode each model generates its own, model-dependent, internal variability. Apart from a direct comparison of MMM results with the observations (Steinbrecht et al., 2017; WMO, 2018; Petropavlovskikh et al., 2019), a comprehensive comparison of the observed changes in the lower stratosphere with observations on timescales from 1998 to present have not yet been performed, and is ~~one of~~ the **main** ~~goals~~ of this study.

From a modelling perspective, averaging multiple CCMs into a MMM suppresses unforced natural variability and therefore reduces uncertainties in trend analyses; it can also lead to a loss of information regarding the sensitivity of CCMs to a changing state, and the range of responses to drivers; warnings against such averaging to understand CCM efficacy have been raised before (Dou-

glass et al., 2012, 2014). Thus, considering the spread in single CCM realisations might provide insight on the probability of the mid-latitude trends occurring by chance, if one or some of the realisations can reproduce the mid-latitude declines. A study investigating the spread of stratospheric ozone trends in nine ensembles members of the WACCM CCM over 1998 to 2016 found trends ranging from ±6% in the lower stratosphere (Stone et al., 2018), a similar magnitude to those in

the observations, though the extremities of this range were only found over the equator, and none of these members showed the spatially-resolved, wide-spread (50°S–50°N) and coherent decreases found in the observations. Absence of coherence in WACCM in the aforementioned ensemble runs does not imply that natural variability is not interfering with trends, but ~~an expansion of exploring~~ **a wider exploration of** this possibility ~~to~~ **across** more models, as we will do here, is needed to build

confidence in this argument.

     Many past studies, and assessments, have usually considered changes in ozone from 1960 to 2100, sub-periods within, or MMM changes since 1998 and 2000 up to the time of the study (Eyring et al., 2010; SPARC/WMO, 2010; WMO, 2014; Dhomse et al., 2018; WMO, 2018). MMM changes in ozone already indicate that by 2013 (WMO, 2014) and 2016 (Petropavlovskikh et al., 2019),

tropical ozone should exhibit negative trends and mid-latitudes positive trends, albeit insignificant in both cases. The recent findings of decreasing lower stratospheric ozone across mid-latitudes and the tropics raises the question of whether any FR models among the MMM can reproduce these changes, and focuses a comparison of FR CCMs specifically over 1985–2016 (or similar periods) with which to compare with recent observational studies. **As such, while the CCMVal-2 report**

**provides an extensive comparison of the models with observations, across multiple timescales and metrics (including, e.g., transport, heating rates, radiative transfer codes, and boundary conditions; see chapter 3 of SPARC/WMO (2010)), ozone trends over the 1985-2016 period were not. Here we consider the specific issue of recent ozone trends over this period** ~~This study will also consider this issue~~.

We find that, to understand the differences (and agreement) between the observations and CCMs, we need to look beyond ozone and determine if the signature of decreasing ozone is consistent with other variables, such as dynamical changes and temperature. More explicitly, the implication of increasing ozone at mid-latitudes in FR CCMs suggest that temperature, for which ozone is a primary driver in this region, might be increasing. Yet, a recent comparison of FR CCMs with improved

lower stratospheric temperature observations showed temperatures have continued to decline in both observations and FR CCMs (Maycock et al., 2018), albeit slower after 2000 than before; while $CO_2$ is responsible for ongoing temperature decreases in the upper stratosphere, it has little influence in the lower stratosphere (Brasseur and Solomon, 2005). As such, the agreement leads to a paradox with respect to ozone and temperature at mid-latitudes that we also resolve here by considering

trends in stratospheric water vapour (SWV), which is also an important driver of trends in the lower stratosphere.

In the following, we first lay-out the suite of ozone, temperature, and SWV observations, reanalysis products for estimates of dynamical changes (section 2.1), and the CCMs we consider (section 2.2). We use dynamical linear modelling (DLM) to estimate long-term changes, and how they

evolve, and fixed dynamical heating (FDH) calculations to quantify temperature changes induced by changes in ozone and SWV; these methods are laid out in sections 2.3 and 2.5, respectively. Following that, we begin by presenting results of changes since 1998 by comparing ozone observations with CCMs in different regions of the lower stratosphere (section 3.1). We use dynamical changes from reanalyses to understand why ozone is decreasing in the tropics and mid-latitudes (section 3.2).

Given the paradox of temperature and ozone changes (section 3.3), we then turn to SWV changes and FDH calculations to assess the importance of radiative processes in the modeled and observed temperature changes (section 3.4). We bring together all of these results in the discussion (section 3.5), and then conclude (section 4).

## 2   Data and methods

### 160   2.1   Observations and reanalyses

For the resolved stratosphere and partial column ozone (PCO), we use the $BASIC_{SG}$ composite as used in Ball et al. (2018) – data are found at https://data.mendeley.com/datasets/2mgx2xzzpk/2 (Alsing and Ball, 2017). This composite merges SWOOSH (Davis et al., 2016) and GOZCARDS (Froidevaux et al., 2015, 2019) ozone composites using the BASIC approach (Ball et al., 2017);

BASIC uses information in both composites to remove artefacts, including jumps and drifts (see examples in Supplementary Materials of Ball et al. (2018)).

For total column ozone, we use SBUV MOD v8.6 (Frith et al., 2014) which shows good agreement with other TCO composites (Chehade et al., 2014; Weber et al., 2018).

Long-term stratospheric temperature observations are limited to a few stratospheric levels, with

particularly low vertical resolution in the lower stratosphere. We use NOAA microwave sounding unit-4 (MSU4) for observations of lower stratospheric temperature; this has a large vertical kernel that peaks at approximately 80 hPa (~18 km) but reaches down to 300 (8-15 km) and up to 20 hPa (~27 km), though the bulk of the kernel is in the stratosphere, roughly between 50 and 150 hPa (Penckwitt et al., 2015).

Stratospheric water vapour (SWV) observational changes are estimated from the filled SWV product of SWOOSH (Davis et al., 2016).

We use the Japanese 55-year reanalysis (JRA-55) (Ebita et al., 2011; Kobayashi et al., 2015) and the Interim European Centre for Medium-Range Weather reanalysis (ERA-Interim; Dee et al. (2011)) fields to investigate residual circulation upwelling ($w^*$) and mixing efficiency, estimated as

the effective diffusivity computed from potential vorticity (Abalos et al., 2016; Haynes and Shuckburgh, 2000)

## 2.2 CCMVal-2 models

We use the REF-B2 CCM simulations from the chemistry climate model validation (CCMVal-2) (SPARC/WMO, 2010; Eyring et al., 2010) as used in the WMO 2014 ozone assessment report to compare with observations. **We note that REF-B2 is not necessarily the optimal scenario of CCM data with which to do comparisons with observations, as it is used for long-term future projections without consistently including external and/or sea surface temperature (SST) and sea ice cover (SIC) boundary conditions. Nevertheless it remains appropriate as this category of data allows for a comparison up to 2016 - neither CCMVal2 REF-B1/B2 nor CCMI REF-C1/C2 have historical boundaries conditions that go up to 2016 - and, further, because REF-B2 has been used for future changes in the ozone layer for previous assessments, and the estimated changes in the 2014 report (using CCMVal-2) and 2018 (using CCMI) are similar, these data are a well-used metric for expected ozone layer changes.**

REF-B2 are simulations to 2100 with future scenario ODS (adjusted Scenario A1) and GHG (SRES-A1b) boundary conditions (SPARC/WMO, 2010) and although solar cycle, prescribed QBO, or volcanic aerosols should be included, they are not included consistently in every case. SST and SIC are provided as boundary conditions from simulations of other climate models such that, e.g., El Nino-Southern Oscillation (ENSO), the major driver of atmospheric variability, does not always resemble observations in the CCMs **(see supplementary materials Fig. S1, and discussion in section 3.5)**. As such, we did not include the aforementioned regressors in the DLM analysis, meaning for CCMVal-2 models, we only derive seasonal cycle and non-linear trends, while for the observations we do (see DLM section 2.3). **We performed a sensitivity test on the observations by applying DLM with and without regressors (Fig. S2) to test the impact on the trend. We found that the trend estimate does not change much between the two cases, although the uncertainties usually increase when no regressors are used**.

We used results from 13 CCMVal-2 models (and a total of 22 ensemble members) as follows. Ensemble means are estimated where more than one exists (number of ensembles in brackets): CAM3.5 (1), CCSRNIES (1), CMAM (3), CNRM-ACM (1), LMDZ (1), MRI (2), Niwa-SOCOL (1), SO-COLv3 (3), ULAQ (3), UMSLIMCAT (1), **UMUKCA-METO (1),** UMUKCA-UCAM (1), and WACCM-CESM (3). We calculated two multi model means (MMMs) including all models (MMM-Am), and a sensitivity including the first ensemble of each (MMM-1m); the results changed little **(and are included in some figures)**. We also checked the sensitivity of the results by excluding CAM3.5 since results in the upper stratosphere (above 20 hPa) are not available; Fig. S5a shows virtually no effect on the middle and lower stratospheric ozone changes. Further, we performed another sensitivity test to see how removal of several CCMs would impact the lower stratosphere, which were chosen due to specific features of the run or output that made ~~testing~~ the impact of their removal on the MMM worth checking. These models were CAM3.5 (~~missing results~~ **no data** in

the upper stratosphere), UMUKCA-METO and UMUKCA-UCAM (climatological SWV); results remained similar, so we do not remove them for the full analysis performed ~~in the paper~~ **here, except as specified**. ~~As no SWV is available for CNRM-ACM and UMSLIMCAT, these are absent in the SWV MMMs and SWV 1998–2016 changes.~~ UMUKCA-UCAM **and UMUKCA-METO** SWV are climatological and display no change and are not presented in the analysis of SWV in the lower stratosphere (Figs. 2, S3, and S5), but are included in the MMMs. Analysis of multi-model means (MMMs) were performed by averaging original model outputs and then performing the DLM analysis.

### 2.3 Regression analysis with dynamical linear modelling (DLM)

Regression analysis is performed using DLM (Alsing, 2019) following Ball et al. (2017, 2018, 2019). Similar to ordinary least squares multiple linear regression (MLR; e.g. WMO (2006, 2014); Harris et al. (2015); Steinbrecht et al. (2017); Ball et al. (2017)) a set of regressors (predictor variables) are used to represent known variability: the 30 cm solar radio flux (F30) (Dudok de Wit et al., 2014), a latitude-dependent stratospheric aerosol optical depth (SAOD; (Thomason et al., 2017)), the NOAA El Nino Southern Oscillation (ENSO) 3.4 index (from NOAA: http://www.esrl.noaa.gov/psd/enso/mei/table.html), two Quasi-Biennial Oscillation (QBO) proxies at 30 and 50 hPa from the Freie Universitaet (http://www.geo.fu-berlin.de/en/met/ag/strat/produkte/qbo/index.html), the Arctic and Antarctic Oscillation, AO/AAO (http://www.cpc.ncep.noaa.gov/products/precip/CWlink/), as proxies for Northern and Southern surface pressure variability, and an auto-regressive (AR1) process (Tiao et al., 1990). In contrast to MLR, the main advantage of DLM is the non-linear trend and evolving seasonal cycle. For the seasonal cycle, DLM estimates 6- and 12- month harmonics for the seasonal cycle at the same time as the other regressor amplitudes. Additionally, the trend is not predetermined with a linear or piece-wise linear model, but is allowed to slowly vary and the degree of trend non-linearity is an additional free parameter that is jointly inferred from the data along with the trend, seasonal cycle, regressor amplitudes, and the AR process; see Laine et al. (2014)] and Ball et al. (2019) for more details. **We do not use regressors for the CCM analysis and a sensitivity analysis using the observations indicates little change to mean trend estimates (see section 2.2.**

### 2.4 Statistics

We infer posterior distributions on the non-linear trends by Markov Chain Monte Carlo (MCMC) sampling using the public code DLMMC (Alsing, 2019). DLM analyses like the one performed here typically have more conservative uncertainties on the trend than MLR since DLM represents a more flexible regression model, and (in this case) formally marginalizes over uncertainties in the regression coefficients, seasonal cycle, autoregressive process and coefficients, and parameters characterizing the degree of non-linearity in the trend (Ball et al., 2019). Probabilities of changes are estimated from the sampled posterior distributions; we apply Gaussian kernel-density estimates (KDEs)

to the MCMC samples to estimate the marginal posterior probability density functions (PDFs), and probabilities of a change quoted in the manuscript are estimated from integrals of these PDFs.

## 2.5 Fixed dynamical heating (FDH) calculations

We use the Parallel Offline Radiative Transfer (PORT) model (Conley et al., 2013) to quantify the (radiative) contribution of ozone and SWV to temperature changes in the stratosphere in models and observations. This is done by imposing ozone and SWV perturbations in PORT, and allowing the stratosphere to radiatively adjust in offline calculations, while keeping dynamical heating and tropospheric temperatures fixed: this is the so-called Fixed Dynamical Heating (FDH) approximation, a method commonly used to compute the stratosphere-adjusted radiative forcing (e.g. Fels et al. (1980)). Following the approach of previous work (Forster and Shine, 1997), we consider the temperature adjustment above the tropopause layer that is required for the stratosphere to reach radiative equilibrium, as the contribution of each of the species to the trends. As not all of the spatial data was available for UMSLIMCAT SWV and CAM3.5 ozone, these are absent in the MMMs for the FDH calculations.

## 3 Results

### 3.1 Ozone: observed mid-latitude lower stratospheric trends do not match modelled changes

The successful implementation of the Montreal Protocol led to TCO depletion halting in ~2000, but no significant increase has yet been observed (Fig. 1a) (Weber et al., 2018; Chipperfield et al., 2018; WMO, 2018). The MMM of 13 CCMs from CCMVal-2 (SPARC/WMO, 2010; WMO, 2014; Dhomse et al., 2018) indicates a significant recovery should be underway (Fig. 1b); all individual CCMs, except one, reflect this behaviour in TCO (Fig. S3a). The 60°S–60°N ozone layer is observed to have likely continued to thin due to lower stratospheric ozone decreases (Fig. 1a) that counteract an upper stratospheric recovery (Ball et al., 2018), which are not reproduced by the MMM (Fig. 1b). While lower stratospheric ozone (Fig. 1a) - **defined as 147-32 hPa in the mid-latitudes, 60°–30°, and 100-32 hPa in the 'tropics', 30°S–30°N** - exhibits a monotonic decline in contrast to the behaviour of TCO, the trends are qualitatively similar in their second derivative (acceleration; Fig. 1c), with a slower post-1997 decline that accelerates after 2009, and similar inflection times after 2000. This correlated behaviour can be explained by the large contribution of the lower stratosphere to the TCO. The same qualitative similarities in TCO and lower stratospheric ozone trends is seen for the MMM, but with acceleration five-times larger compared to the observations (Fig. 1d). Nevertheless, observation-model lower stratospheric ozone changes disagree significantly (Fig. 1a,b) and drive much of the TCO observation-MMM difference, although it should be noted that uncertainty remains in changes within the tropospheric component of TCO (Ball et al., 2018; Gaudel et al., 2018; Ziemke et al., 2018). We note that the 50°–60° region in both hemispheres shows relatively

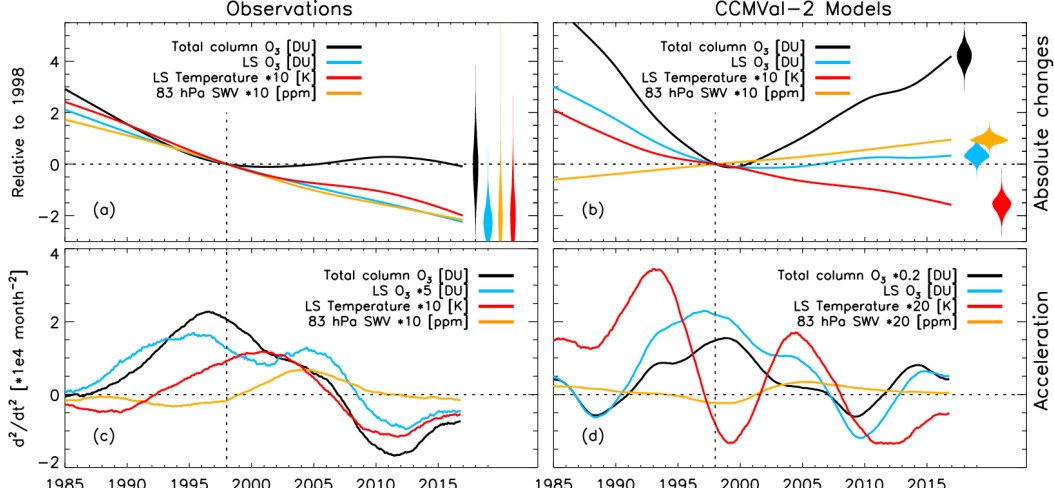

Figure 1: Global 60°S–60°N 1985–2016 stratospheric changes. **(a)** Observed non-linear trends for total column ozone (black), lower stratospheric ozone (blue), temperature (red) and stratospheric water vapour (SWV, yellow) relative to 1998, and their respective **(c)** acceleration curves (5–year smoothing); **(b,d)** as for (a,c) but for the multi-model mean (MMM). Units and scaling of each variable are indicated in the legends.

flat lower stratospheric ozone trends (Ball et al., 2019), and therefore the quasi-global integrated changes are driven by the 50°S–50°N region (see similar results in Figs. S3–S5); we therefore focus on this region.

Figure 2a–c shows the observed, **individual** CCM ensemble **members** and MMM changes in lower stratospheric ozone from 1998–2016, in three sub-regions: southern hemisphere mid-latitudes (SH, 50°–30°S), the tropics (20°S–20°N), and northern hemisphere mid-latitudes (NH; 30°–50°N). Total column ozone (Fig. S4) and quasi-global (60°S–60°N, Fig. S3; 50°S–50°N, Fig. S5) changes are provided in the supplementary materials. A MMM sensitivity test that considers only one-ensemble member of each model (MMM-1m) to avoid biasing the MMM to models with more members, shows little difference to including all (MMM-Am). Over the tropics (Fig. 2b), both the MMM and observations indicate a significant decrease and, while some CCMs agree in the magnitude, observations show a stronger decrease than the MMM. At mid-latitudes (Figs. 2a, c), however, the MMM indicates a significant increase, while observations show a decrease. It is this opposing behaviour at mid-latitudes, and the smaller MMM decrease in the tropics, that leads to the opposing trends in the integrated quasi-global lower stratospheric ozone (Fig. 1a, c). We therefore need to consider the equatorial and mid-latitude changes separately.

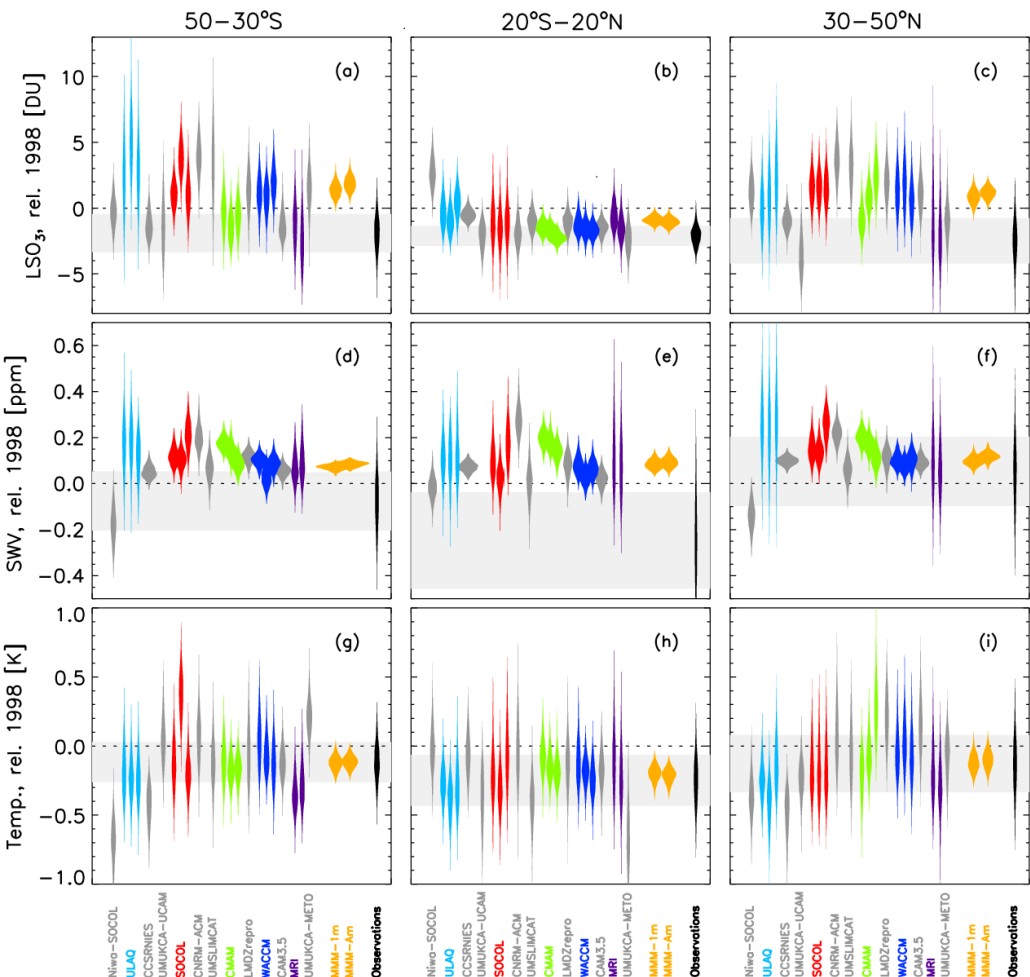

Figure 2: Lower stratospheric 1998–2016 ozone, water vapour, and temperature changes in models and observations. **(Left column)** 50–30°S, **(middle)** 20°S–20°N, and **(right)** 30–50°N; **(Upper row, a–c)** partial column ozone (147–32 hPa, 50–30°S/30–50°N); 100–32 hPa 20°N–20°S), **(middle, d–f)** stratospheric water vapour (SWV) at 83 hPa; **(lower, g–i)** temperature estimated from the MSU4 observing kernel. Violins represent double-sided probability distribution functions. Observations are black (right) with grey-bands representing the 68% highest density (most likely) interval; models are grey (single member ensembles) and colours are for models with more than one ensemble.

### 3.2 Dynamics: evidence for increased tropical upwelling and mid-latitude mixing

The decrease in tropical ozone shown by most CCMs can be explained by an increase in tropical residual upwelling; upwelling is inversely correlated with tropical ozone over 1960–2100 in CCMVal-2 simulations (Fig. S11 of Eyring et al. (2010) and Fig. 9.6 of SPARC/WMO (2010)). It is well-established that later in the 21$^{st}$ Century a decline in tropical lower stratospheric ozone should emerge due to an acceleration of the large-scale Brewer-Dobson circulation (BDC) (Hardiman et al.,

2014; Butchart, 2014). This tropical lower stratospheric ozone decrease is actually already apparent in the spatially resolved changes presented in Fig. 3 in most CCMs. The magnitude of change is smaller in the MMM (Fig. 3b; see also WMO (2014) considering 2000–2013) compared to most of the individual CCMs (Fig. 3c–k), and observations (Fig. 3a) (WMO, 2014). The reason for a smaller tropical lower stratosphere ozone MMM decrease is because the magnitude and position of

maximum decrease varies by CCM, and Niwa-SOCOL and ULAQ even show opposing (i.e. positive) ozone changes (Fig. 3m–n). Overall, the implication is that part of the observed tropical lower stratospheric ozone decrease over 1998-2016 is likely to be driven by an acceleration of the BDC.

To determine whether a BDC acceleration is indeed driving the ~~the~~ lower stratospheric ozone decrease, we analyse 1998–2017 upwelling changes in two reanalysis products (JRA-55 (Ebita et al.,

2011) and ERA-Interim (Dee et al., 2011)), which represent observed historical changes in the circulation, at pressure levels just above the tropopause in Fig. 4b. We see an increase in residual upwelling at 96 hPa, which is highly likely (≥98% probability) in both reanalyses and **at least** ~~approximately~~ ~~five~~ two times larger in magnitude than the CCMVal-2 **MMM, although some models imply similar changes** ~~(not shown)~~. At 80 and 67 hPa we see a likely (>90%) residual upwelling

increase in JRA-55, while ERA-Interim shows decreasing confidence with height**; the CCMs agree better with JRA-55 at these two levels than ERA-Interim, especially at 67 hPa**. The 1998-2017 timeseries is short compared with the large interannual variability; using longer timeseries (1979–2017) to better constrain regressors does not change the conclusions. Therefore, our results provide evidence that enhanced upwelling, likely related to GHGs, i.e. climate change, has already been driv-

ing a tropical ozone decrease over 1998–2017 in both CCMs (Eyring et al., 2010; SPARC/WMO, 2010; Polvani et al., 2018, 2017) and observations (Ball et al., 2018, 2019).

At mid-latitudes (30-50°N/S), three CCMs display some decrease over the 1998-2016 period (Fig. 3). Notably, UMUKCA-UCAM, and MRI display mid-latitude decreases (Fig. 2a,c) and spatial patterns (Fig. 3j,k) most reminiscent of the observations; **UMUKCA-METO shows a decrease**

**only in the NH lower stratosphere (Figs. 2c, 3o)**. Nevertheless, eight other CCMs suggest mid-latitude ozone increases consistent with enhanced downwelling in the shallow branch of the BDC. These differences at mid-latitudes lead to the MMM and observations disagreeing in the quasi-global mean. To understand this discrepancy, we turn to other lower stratospheric variables.

It has been recently noted that the negative ozone trends in the lower stratosphere may be a result

of enhanced isentropic mixing between the tropics and mid-latitudes, based on MERRA-2 reanalysis

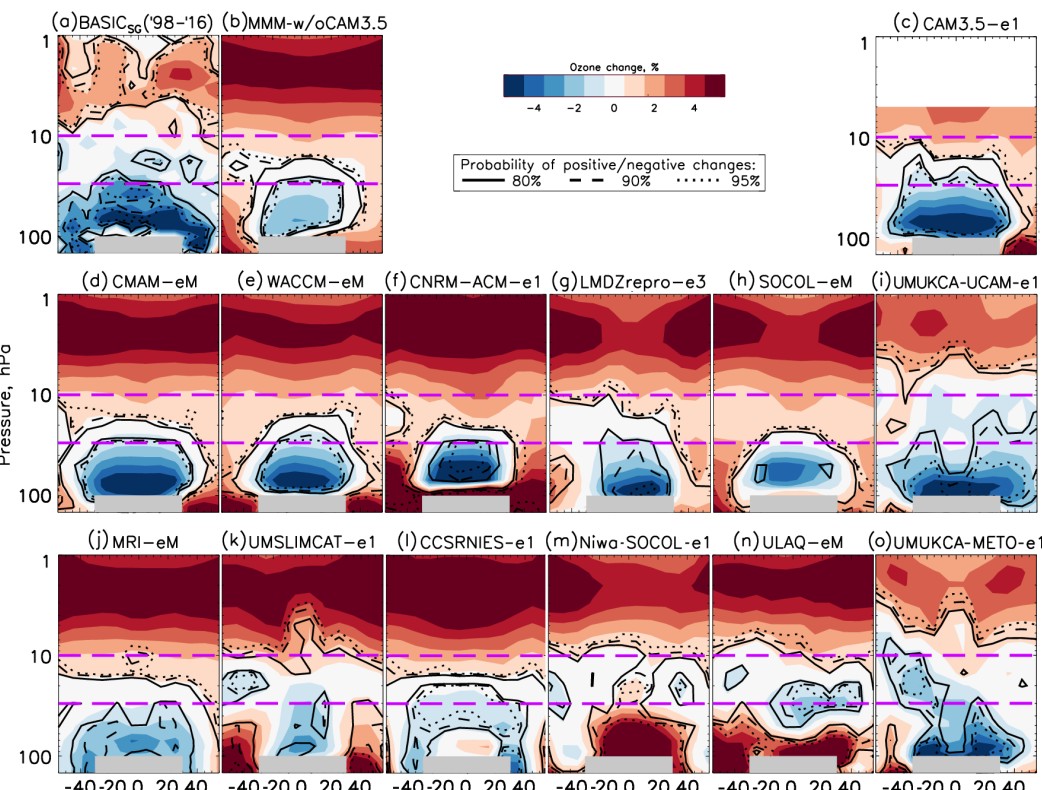

Figure 3: Latitude-pressure ozone changes from 1998–2016. **(a)** Observations, **(b)** CCMVal-2 MMM without CAM3.5, and **(c–n)** ensemble mean (eM) and single ensemble members (e1/e3) from each CCMVal-2 model. Colours represent positive (red) and negative (blue) changes (upper legend); contours represent probabilities of a positive or negative change (lower legend); grey shading represents the tropical troposphere, which is omitted. All changes are calculated considering only data from 1998–2016. All individual members of ensemble means are shown in Fig. S6; MMM results including CAM3.5 and a sensitivity test without five models are provided in Fig. S7.

(Wargan et al., 2018), although in that study mixing was not explicitly calculated, whereas we will do so here. Interestingly UMUKCA-METO, similar to UMUKCA-UCAM (differing primarily in how halogen washout and aerosol heating is treated (SPARC/WMO, 2010)), displays much larger mixing efficiency (Dietmüller et al., 2017) than any other CCM[1]**though this does not appear to**
**lead to a larger response in lower stratospheric ozone (Fig. 3)**. MRI also displays above average mixing efficiency relative to other CCMVal-2 models (Dietmüller et al., 2017). Both the large-scale BDC transport and mixing are expected to increase in the future (SPARC/WMO, 2010; Abalos et al., 2017). **This might imply that the MRI, UMUKCA-UCAM, and (NH) UMUKCA-METO mid-**

---

[1]This includes models used in the chemistry climate model initiative phase 1 (CCMI-1 (Morgenstern et al., 2017)), updated since the chemistry climate model validation phase 2 (CCMVal-2 (SPARC/WMO, 2010)) models used here.

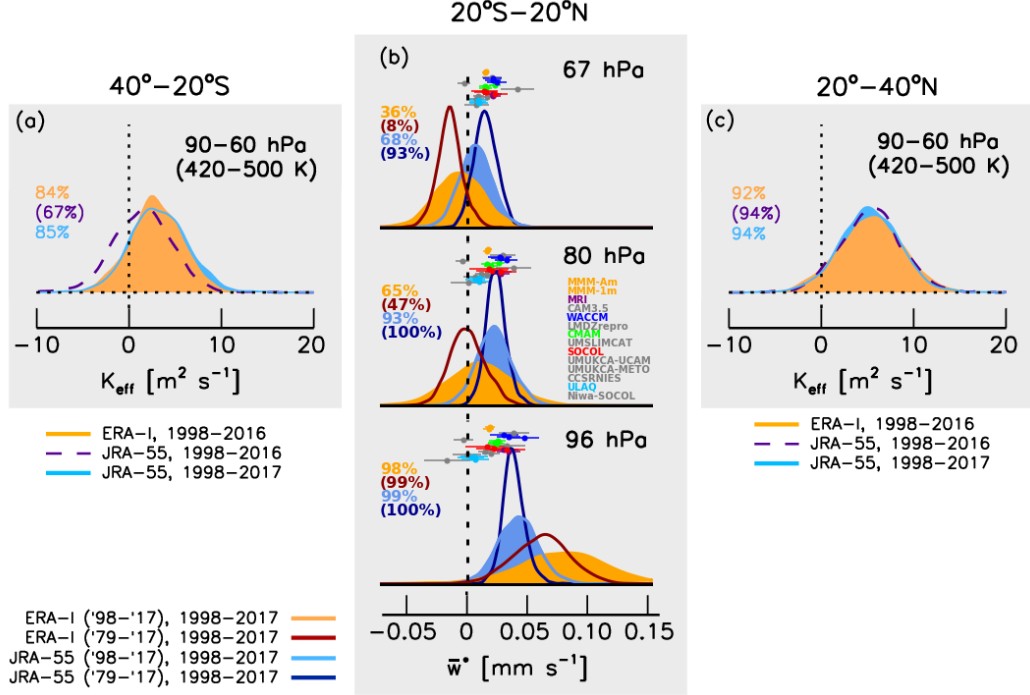

Figure 4: Effective latitudinal mixing and tropical upwelling changes since 1998. **(a)** southern hemisphere latitude-pressure averaged changes in latitudinal mixing, $K_{eff}$ (40–20°S); **(c)** as for (a) but for northern latitudes (20–40°N). **(b)** tropical (20°N–20°S) upwelling changes ($\overline{w}^*$) at three pressure levels: **(top)** 67 hPa, **(middle)** 80 hPa, and **(bottom)** 96 hPa. Estimates are made from reanalysis for periods in the legends; CCM estimates consider data over 1985-2017 with $1\sigma$ error bars. In (b), solid probability distribution functions (PDFs) are for changes estimated using data covering 1998–2017, while line-PDFs are using 1979–2017; for (a) and (c) ERA-Interim has a solid PDF for 1998–2016, while JRA-55 has a solid (line) PDF for 1998–2017 (1998–2016). Percentages are the probability of positive changes in all PDFs; brackets surround percentages for the line PDFs. Timeseries for (b) are provided in Fig. S8; Fig. S9 for (a,c).

**latitude ozone decreases because of higher mixing efficiency in these models, and vice versa for**
**the majority of CCMs, although a recent study by Orbe et al. (2020) indicates that large-scale**
**changes in advective transport may be more important**.

In addition to previous work considering MERRA-2 reanalysis (Wargan et al., 2018), we **add** ~~provide further~~ supporting observational evidence that mixing has increased since 1998 using JRA-55 and ERA-Interim reanalyses. Figs. 4a and 4c indicate that mixing across the sub-tropics between the equator and the SH and NH, respectively, increased over 1998–2016 (and 1998-2017) in both ERA-Interim and JRA-55 reanalyses (estimated from effective diffusivity (Haynes and Shuckburgh, 2000; Abalos et al., 2016) in Fig. S9). The increase in mixing is larger and more probable in the NH (>92%) than the SH (>66%), which is in agreement with the NH displaying larger mid-latitude decreases than the SH (Ball et al., 2018; Chipperfield et al., 2018; Wargan et al., 2018; Ball et al., 2019; Orbe et al., 2020). Thus, ~~there is now consistent~~ observational evidence in support of enhanced mixing to mid-latitudes in the recent past **is consistent across reanalyses**.

### 3.3 Temperature: imprints of decreasing ozone

The aforementioned changes in ozone and transport, if correct, should be found in other stratospheric variables: ozone is not an isolated quantity, and the 1998–2016 reduction in lower stratospheric ozone should lead to reduced radiative heating and a decrease in observed temperature (London, 1980; Brasseur and Solomon, 2005). Quasi-global lower stratospheric temperature from observations (see Methods) is shown in Fig. 1a; the temperature evolution mimics the pre-1998 ozone decreases, flattening through the 2000s, and then continuing to decrease after 2009; the behaviour of the acceleration curve (Fig. 1c) also follows the variations in ozone post-2002, as expected physically. A recent analysis of updated temperature trends (Maycock et al., 2018) concluded that the negative 1998–2016 temperature trend was smaller compared to 1979–1997, as a result of reduced loss of ozone, caused by a phase-out of ODS emissions; the qualitatively consistent ozone and temperature trends (Fig. 1c) supports this conclusion. By 2016, observed quasi-global temperature (60°S–60°N) is approximately 0.20 K lower than in 1998 (Fig. 1a); the same is true for the MMM (0.15 K; Fig. 1b), and across latitude bands (SH, tropical, and NH; Fig. 2g–i) for individual CCMs.

However, while temperature trends are consistent with ozone in the tropics, there are inconsistencies in the mid-latitudes, where MMM and observed 1998–2016 temperature changes agree with observations, but ozone trends do not. To estimate the impact of ozone on temperature trends, we applied the FDH approximation (section 2.5) to the spatially-resolved observed (Fig. 5a) and MMM (Fig. 5b) 1998–2016 ozone changes within a CCM (Fig. 5d,h; see Methods), and then applied the MSU temperature observing kernel to yield the ozone contribution to the temperature decrease (Fig. 5e, j); the MSU4 kernel as presented in Randel et al. (2009b) is plotted in Fig. 5 between panels f and g. We note that FDH provides a first-order estimate of the ozone contribution to temperature changes, as it neglects non-radiative processes such as dynamical adjustments. We find

that the ozone contribution to the observed temperature change, quantified via the FDH approximation, agrees with the observed temperature changes throughout all latitudes (Fig. 5e). Integrated over the 60°S–60°N region, ozone (radiatively) contributes to a temperature change of -0.24 K. The coherent changes in ozone and temperature in observations (Fig. 1a), along with the close match between FDH calculations imposing ozone changes confirm that ozone is the major contributor to

the observed temperature decreases over 1998–2016 (Fig. 5e). The story is different when applying the FDH approximation to the ozone changes in the MMM: as expected, tropical ozone decreases should lead to cooling (Fig. 5j), but the mid-latitude ozone increase is inconsistent with the temperature decrease in the MMM, and the 60°S–60°N quasi-global FDH temperature change induced by ozone is only ~+0.01 K. Therefore, for this to be physically consistent with the MMM 1998–2016

temperature decreases, something else must be driving the lower stratospheric cooling in CCMs.

**3.4 Stratospheric water vapour: reconciling observed and modelled temperature trends**

In addition to ozone, stratospheric temperatures are affected by radiative effects from, and chemical changes in, $CO_2$, $N_2O$, and $CH_4$ (Revell et al., 2012; Portmann et al., 2012; Nowack et al., 2015) and stratospheric water vapor (SWV) (Forster and Shine, 1999; Dessler et al., 2013). While cooling

from $CO_2$ is important in the upper stratosphere, near the tropopause it has little relative contribution (Shine et al., 2003; Brasseur and Solomon, 2005; Maycock et al., 2011). SWV is the next most important contributor to lower stratospheric temperature changes and has the opposite effect on temperature to ozone in the lower stratosphere, i.e. cooling if SWV increases (Shine et al., 2003; Brasseur and Solomon, 2005; Maycock et al., 2011). For the FDH-estimated ozone contribution to

temperature changes to be consistent across latitudes (Fig. 5), SWV in the MMM would need to increase after 1998 (Gettelman et al., 2010), while observed SWV (Davis et al., 2016) needs to change little or decrease slightly by 2016. This is exactly what we find: MMM SWV at 83 hPa (close to the peak of the observing kernel of the MSU4 temperature observations; Fig. 5) increases almost linearly over 1985–2016 (Fig. 1b), while SWV in observations shows a continuous decrease from 1994,

flattening slightly after 2000 (Fig. 1a); the picture is more nuanced across latitude bands (Fig. 2d–f). Observed quasi-global SWV decreases are dominated by the tropics (Fig. 2e), which is also where the discrepancy between SWV trends in the MMM and observations is largest. The observed changes in SWV lead to hemispheric differences in the FDH estimated contribution to temperature (Fig. 5e), with overestimation of the trend at Northern latitudes, and underestimation in the tropics and South-

ern latitudes, although the total FDH estimate, when combined with ozone, remains within the 68% credible intervals. The FDH-estimated SWV contribution to the MMM temperature changes lead to improved agreement with the MMM temperature change and with observations (Fig. 5j); the quasi-global FDH estimate for SWV in the observations and MMM is +0.10 and ~~-0.16~~ **-0.18** K, respectively. The combined quasi-global ozone and SWV contributions to the observed and MMM

temperature changes are in agreement within uncertainties, i.e. -0.14 and ~~-0.15~~ **-0.17** K respectively

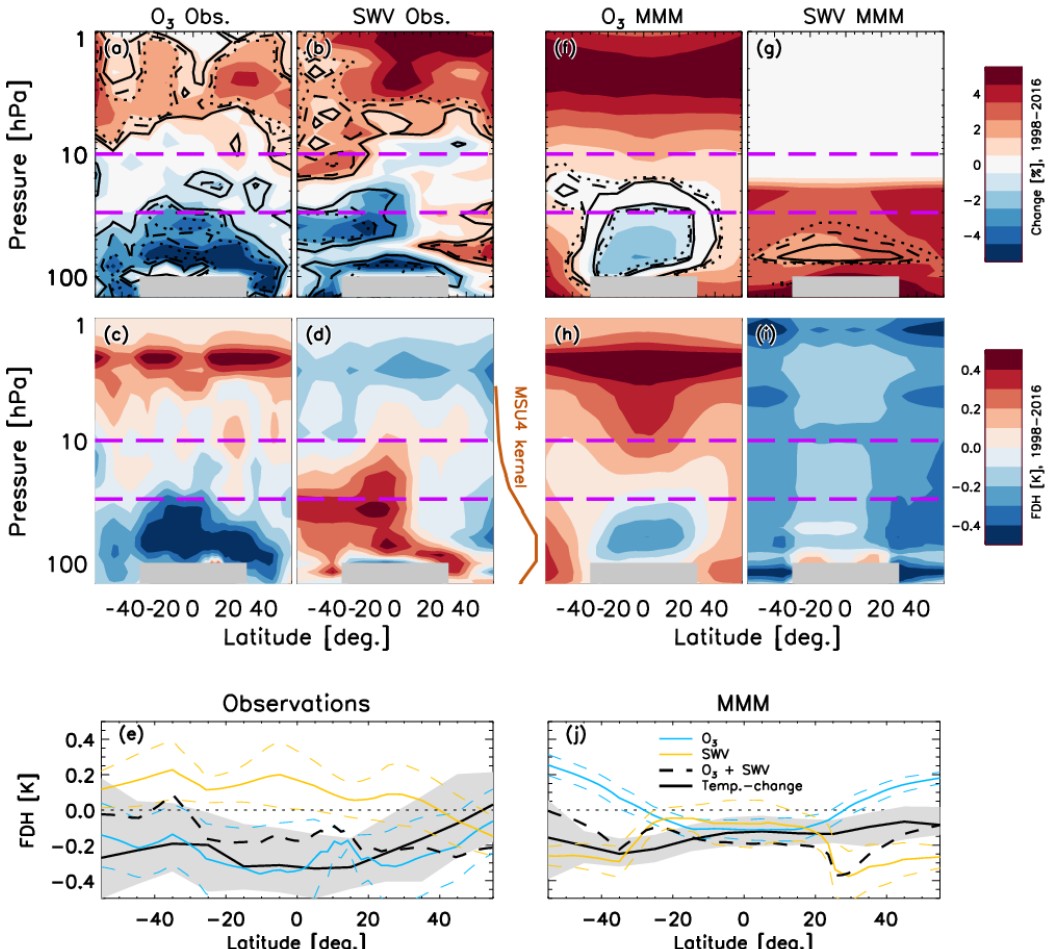

Figure 5: Fixed dynamical heating estimate of ozone and SWV contribution to lower stratosphere temperature changes. **(a–e)** Observed and **(f–j)** MMM estimates for **(a,f)** ozone and **(b,g)** SWV changes (right legend), the corresponding spatially resolved FDH-estimated contributions to temperature changes from **(c,h)** ozone and **(d,i)** SWV temperature; **(e,j)** after applying the MSU4 observing kernel, the estimated latitudinal contribution to MSU4 temperature changes with 68% credible intervals; the MSU4 kernel (Randel et al., 2009b) is plotted between panels d and h.

(Figs. 5e, j), and with the directly observed quasi-global cooling (Fig. 1a). **First, the agreement between FDH and temperature trends indicates that radiative processes largely contribute to the temperature trends in the lower stratosphere. Second,** while the contribution of SWV and ozone to temperature changes over 1998-2016 in the MMM do not agree with observations at mid-latitudes,

their opposing tendencies offset each other and lead to a coincidental agreement in temperature.

SWV changes are not required to explain the observed temperature changes (using FDH, within the uncertainties), but are required to explain the MMM-observations agreement in temperature in spite of opposing ozone trends. The enhanced upwelling should lead to cooling, which is not included in the FDH estimate, and might be a missing component in the difference between the

combined FDH SWV-ozone contribution to the temperature change (Fig. 5e). The difference in FDH-estimated observation-model temperature changes, as well as the larger uncertainties in the FDH-estimate from observations, could be explained by natural variability in the observations that is suppressed in the MMM from averaging natural variability over multiple models. In summary, the temperature changes in the MMM and observations agree fortuitously over the 1998-2016 period,

since the changes in trace gases driving those temperature changes disagree.

### 3.5   Discussion

Bringing together all of the results presented here – ozone, temperature, SWV, upwelling, and mixing – we can hypothesize the likely mechanism driving the long-term changes in the lower stratosphere. Tropical upwelling appears to be increasing (Fig. 4b), and modelling studies indicate this to result

from increased GHGs (Eyring et al., 2010; Polvani et al., 2018) that drive climate change. This directly leads to a decrease in tropical lower stratospheric ozone (Fig. 2b). Further evidence suggests that mixing of air from the ozone-poor tropical lower stratosphere to mid-latitudes has enhanced (Fig. 4a,c), and we **consider this a possible and contributing** ~~conclude that this is a probable~~ cause of the observed ozone decreases at mid-latitudes (Fig. 4a,c). The consequence is that the

continuing ozone decrease is driving the majority of the ongoing temperature decrease in the lower stratosphere at tropical and mid-latitudes (Figs. 2g–i and 5e) (Maycock et al., 2018), as the FDH calculations confirm. Most CCMs reproduce the tropical upwelling and associated ozone decrease (SPARC/WMO, 2010), but CCMs with higher mixing efficiency (Dietmüller et al., 2017) **appear to** produce ozone trends more similar to the observations at mid-latitudes (Figs. 2a-c and 3) **though this**

**is an inference based on a low number of models (2) and may be compensating for a deficiency in large-scale advective transport (Orbe et al., 2020) and requires further consideration**. The role of enhanced mixing in driving ozone trends at mid-latitudes is supported by the observational results estimated from reanalyses (Fig. 4a,c). Further, the temperature decreases in CCMs agree with observations (Fig. 2g–i) because SWV is increasing in the CCMs (Figs. 1 and 2) thus cooling the

mid-latitude lower stratosphere (Fig. 5j); observations show no confident change in SWV at mid-

latitudes (Fig. 2d,f), though we do not have an explanation as to why modelled SWV changes do not agree with observations.

However, many caveats and open questions remain. **We point out that a MMM does not necessarily provide physically meaningful insights (SPARC/WMO, 2010) and may provide confidence in CCMs that show similar, e.g., trends for different reasons. That said, a MMM does provide an aggregate metric for the general behaviour for a group of CCMs when individual CCMs are not downgraded or removed for their poor performance, with the assumption that the influence of poor physical representation is diminished through the act of averaging.** The mechanism proposed here – with SWV and ozone driving the majority of temperature changes – does not fully explain the different changes in temperature between each CCM (Fig. 2); this will require a deeper, case by case examination of how each model is operating. The CCMs considered here are a part of the CCMVal-2 model intercomparison that preceeds the more recent CCMI-1, but nevertheless other studies have shown that results between CCMVal-2 and CCMI-1 are consistent in their multi-decadal changes in SWV (Smalley et al., 2017), ozone (Dhomse et al., 2018), temperature (Maycock et al., 2018), and with upwelling and mixing (Dietmüller et al., 2017), and are therefore still representative of the state-of-the-art. **Nevertheless, large-scale CCM transport deficiencies exist in most models, such that while there is consistency across models, comparisons across multiple metrics indicate shortcomings in transport, e.g. even in the representation of seasonal cycle variability in southern hemisphere lower stratosphere transport (SPARC/WMO, 2010).**

The CCM simulations analyzed in this study also mainly consider long-term changes in ODSs and GHGs, but do not prescribe the observed SSTs, which means natural variability in temperature is likely different to that of the observed world. As such, the effect of large natural variability affecting the temperature trend estimates is not taken into account in this study (Ball et al., 2019); whether natural variability or the GHG forcing signal is underestimated in the CCMs, and is the cause of the difference with observations, remains an open question. **One important aspect of the analysis performed here is that the CCMs do not include regressor terms, due the absence of information to make fair comparisons when using different sets of regressors, and since observations with and without regressors display similar mean trends (Fig. S2), this implies that the length of the timeseries is long enough to mitigate the effect of short-term behaviour from forcing agents such as ENSO on the trend estimates. Indeed, SSTs are expected to have a large impact on stratospheric variability, usually represented by ENSO variability (Randel et al., 2009a; Calvo et al., 2010). But our results appear to indicate that, at least for the model scenario (REF-B2) considered here, SSTs do not have a clear impact on the trend estimates, and it is likely that other factors have a more significant impact. For example, the range of trend estimates in SH lower stratospheric temperature between SOCOL ensemble members is as large as the range between all other models (Fig. 2g), despite using the same SST forcing (see Fig. S1 and SPARC/WMO (2010)), while the set of other CCMs use seven other varieties of SST boundary**

**conditions (Fig. S1). Similarly a large range of changes can found between ULAQ, WACCM and CAM3.5 that all use the CCSM3 SST as a boundary condition. Counterexamples can also**
**be found, but there is little consistency between the relative trend estimates of CCMs across variables (i.e. Fig. 2) depending on the SST boundary conditions.**

**So, the overall implication is that SST boundary conditions cannot be singled out as a major factor influencing the trend estimates, when other aspects of (atmospheric) internal variability or between different CCM design appear to be responsible for a similar or, more likely, larger**
**impact on the stratospheric variability. For example, the Chemistry Climate Model Validation phase 2 (CCMVal-2) report provides an extensive intercomparison and discussion of the deficiencies across CCMs in simulating transport (chapter 5), particularly at around 100 hPa in the lower stratosphere, and the modelling of the QBO was considered 'too primitive' to make an assessment at that time (chapter 8) and is an issue needing further work, especially with**
**respect to its impact on modelled ozone. While focus in these examples was on variability, it is not surprising that trends may consequently differ too. As such, untangling and identifying the aspects responsible for the spread in trends remains an important focus in model evaluation.**

We note that the potential for internal variability to bias trends ~~We note that this~~ has been discussed in several recent studies (Ball et al., 2018; Chipperfield et al., 2018; Wargan et al., 2018;
Stone et al., 2018), but an update in ozone trends shows that the observed negative lower stratospheric ozone trends persist despite large interannual variability (Ball et al., 2019). Further, one might expect decreasing temperatures near the tropical tropopause entry point to freeze-out more water vapour from air entering the lower stratosphere. However, this is not what CCMs show, and the temperature changes at the entry point is hard to predict due to stratospheric cooling, tropo-
spheric warming, and a rise of the tropopause (Gettelman et al., 2009; WMO, 2018), as well as other processes such as convective over-shooting and isentropic mixing with mid-latitudes complicating the picture further. The large altitude range of the MSU-4 kernels applied to the CCMs that includes the upper troposphere may hide a rising or warming tropopause region (SPARC/WMO, 2010), inhibiting attribution to the cause. Numerical diffusion in CCMs might also allow water vapour to
incorrectly enter the lower stratosphere in models and should also be considered for further evaluation. Finally, multi-decadal (natural) variability is an alternative hypothesis to the signals presented here being climate-change driven, although a specific internal driver to attribute the signal is not currently available, so GHG increases remain, in our view, the more likely hypothesis at this stage.

## 4 Conclusions

In summary, we have presented results that show the behaviour of decreasing ozone in the lower stratosphere appears to be imprinted on temperature changes and might be explained by enhanced upwelling and increased horizontal mixing; at least part of the tropical changes can be attributed

through models to an acceleration of the BDC due to rising GHGs (SPARC/WMO, 2010; Polvani et al., 2018). Tropospheric temperature increases due to increased GHG emissions modify the thermal wind balance and strengthen the sub-tropical jets in the lower stratosphere, which subsequently affect wave dissipation (Garcia and Randel, 2008; Shepherd and McLandress, 2011) that directly influences the strength of upwelling and mixing (Wargan et al., 2018) in the lower stratosphere. If ozone decreases in the tropical lower stratosphere and then mixing and transport to mid-latitudes is enhancing, as we indeed find, a decrease in ozone both in the tropics and mid-latitudes is the expected, and observed, outcome (Ball et al., 2018). Our results suggest that the quasi-global lower stratospheric ozone decline can be explained by climate-change related changes in transport and mixing in the lower stratosphere.

However, confidence in future projections using CCMs relies on agreement with observations over the historical record; indeed, the two CCMs displaying mid-latitude decreases (MRI and UMUKCA-UCAM) do project a mid-latitude recovery by the middle of this century **(Fig. S10)**. However, since we do not yet know why CCMs in general do not reproduce the observed ozone decreases in the mid-latitudes, or indeed why these two do, open questions remain about the future of lower stratospheric ozone and the ozone layer under a changing climate.

*Author contributions.* G.C. prepared the model data, W.T.B prepared the observational data, and W.T.B. and J.A. performed the DLM analysis; W.T.B and J.A. did the statistical analysis. G.C. performed the FDH calculations. M.A. prepared ERA-Interim and JRA-55 reanalyses, and calculated mixing and upwelling variables. W.T.B. prepared figures and wrote the manuscript. All authors contributed to the manuscript.

*Acknowledgements.* W.T.B. was funded by the SNSF projects 200020_163206 (SIMA) and 200020_182239 (POLE). G.C. is funded by the SNSF Ambizione grant PZ00P2-180043. 'BASIC$_{SG}$' for 1985–2016 is available from https://data.mendeley.com/datasets/2mgx2xzzpk/2. CCMVal-2 model data are from the Stratospheric Processes And their Role in Climate; Eyring, V. (2012): CCMVal-2 (Chemistry Climate Model Validation Activity 2) coupled chemistry climate models outputs located at the NCAS British Atmospheric Data Centre, available at http://data.ceda.ac.uk/badc/ccmval/data/CCMVal-2, accessed 2020/05/27. The DLM algorithm (Alsing, 2019) is available at https://github.com/justinalsing/dlmmc. N3.4 data was last accessed on 6 June 2020 at https://psl.noaa.gov/data/correlation/nina34.data (Trenberth, 2020).

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
