# Peer review of "Inconsistencies between chemistry climate model and observed lower stratospheric ozone trends since 1998"

_Atmospheric Chemistry and Physics, 2019_

## Referee Comment (RC1) · Anonymous Referee #3 · 24 Oct 2019

The motivation for this study is as follows: While CCMs show tropical O3 decline over the past 20 years that is likely driven by increases in tropical upwelling, models do not produce the (observed) decline in midlatitude O3; rather, they show an increase. The authors use a fixed dynamical heating model to estimate the impact of the negative O3 trend on the temperature trend. The result is that the observed temperature trend is consistent with the observed O3 trend. The conundrum the authors find is that in spite of the disagreement between model and obs midlatitude O3 trends, the models and obs get similar temperature trends – this is not the physically expected response. The authors propose that the explanation for this is the models have stratospheric water vapor trends that are opposite to those observed, thus creating a dynamical heating term

that opposes that of the simulated O3 trend, leading to fortuitous agreement with the observed temperature trends. A sort of 'two wrongs make a right'. This lack of agreement between the CCMVal-2 models and observations motivates this study, whose intent is to explain what's wrong with the models and thereby help improve them and increase confidence their O3 projections.

Much of the knowledge regarding model problems that is so 'urgently needed' is already available in the SPARC (2010) CCMVal evaluation. It has over 400 pages of detailed analyses showing why these models don't match observed O3, temperature, trace gases, variability and more. Chapter 2 is very useful because it details how each model differs in its representation of radiative, chemistry, and dynamical processes, boundary conditions, etc. Chapter 3 is all about radiative processes and the information presented reveals much about how these models can be expected to respond to changes in radiatively active trace gases, i.e., O3. Chapter 5 reports on the transport issues affecting the credibility of their ozone simulations, in particular how well they represent tropical ascent and mixing out of the subtropics – topics so very relevant to how models' circulations will be respond to increasing GHGs and alter future ozone distributions. Chapter 8 examines simulated O3 variability and whether models have the necessary processes to simulate that variability (spoiler alert: they don't). This report may be 9 years old but there is much about models and the physical processes requisite for simulating ozone that must be understood before undertaking an investigation of why CCMs don't do a particular thing right.

The authors justify their use of the CCMVal2 simulations rather than the CCMI runs because these models (in some ways?) are 'still representative of the state-of-the-art'. I think it is a mistake not to use the CCMI runs because although some models may be the same as in CCMVal2, others have made improvements. (Shouldn't the goal be to improve the current state of modeling?) Whether you use CCMVal2 or CCMI simulations, the conclusions of the SPARC (2010) report still provide a relevant starting point for a study like this.

Using these CCMs' transport behavior to support the interpretation of observed O3 trends in terms of ascent and mixing is problematic given the many model transport problems diagnosed in Ch. 5 of the SPARC report. See in particular Figure 5.20 that evaluates tropical ascent and meridional mixing. Most of the models used in this paper did rather poorly here. When a model does a poor job at representing a physical process, it should never be assumed that the simulated process will (magically) respond in a physically meaningful way to changes in forcing (e.g., increasing GHGs).

Furthermore, given that these CCMs have many different radiative, chemistry, and dynamical problems, using the multi-model mean (MMM) is a bad idea. The MMM is a mash-up of correctly and incorrectly simulated (or missing) processes. This manuscript works from the assumption that something physically meaningful can be derived from their analysis of the MMM. That's not possible. As an example, here are some relevant conclusions from the Ch. 3 summary of the models' radiation evaluation that speak to what you get with a MMM:

". . .5 out of 18 CCMs show biases in their [temperature] climatology that likely indicate problems with their radiative transfer codes." "Problems remain simulating radiative forcing for stratospheric water vapour and ozone changes with a range of errors between 3% and 200% compared to LBL [line by line] models." "The stratospheric water vapour forcing has errors of over 100% between the models (Figure 3.12)."

Did you know water vapour in some of the models can't respond to climate change because a water vapour climatology was used? I also read in the report of a situation where the MMM quantity (I don't remember which one) actually got a higher grade than the models did individually on a particular evaluation. Bottom line: the MMM is not a quantity from which you can derive physical meaning.

I'm curious why the authors excluded 6 of the 18 CCMs models for the MMM. The excluded models (AMTRAC, EMAC, E39, GEOSCCM, UMetrac, and UMUKCA-Meto) all provided O3, H2O, and T outputs.

[Figure]

On a different topic, the authors report that lower stratospheric ozone has negative trends, a conclusion not broadly agreed upon by the community. See for example Steinbrecht et al. (2018), WMO (2019), and the LOTUS report (2019). None of these publications finds statistically significant O3 trends below the upper stratosphere. Steinbrecht et al. note that the search for ozone trends is complicated by ozone variations not caused by declining ODSs, such as variability in or changes to the Brewer-Dobson circulation. The role of circulation variability on ozone trends is acknowledged by the authors, yet they seem not to recognize that the CCMs' inabilities to provide anything close to observed interannual variability in stratospheric composition is one of the greatest shortcomings that interferes with the ability to simulate credible O3 projections. The QBO is the greatest source of stratospheric composition variability and these models either have no QBO or an unrealistic one. This is the elephant in the room with respect to 'why don't the CCMs get a negative or flat O3 trend over the past 20 years'. See Chapters 4, 5, and 8 in the SPARC report.

The overarching motivation of this paper – to improve the credibility of CCMs– is fine. The analysis of individual model behavior, especially in a model with vetted radiative, chemical, and transport processes, may produce useful insights into CCM needs. However, the problem stated by the paper's title – inconsistencies between chemistry climate model [sic] and observed lower stratospheric trends – cannot be solved with the chosen approach. The results presented rely on analyses of the physically meaningless MMM, which is fundamentally not a valid approach; for this reason I do not recommend publication.

---

## Referee Comment (RC2) · Anonymous Referee #1 · 27 Oct 2019

The paper 'Inconsistencies between chemistry climate model and observed lower stratospheric trends since 1998' by Ball et al. discusses recent ozone, stratospheric water vapor (SVW) and temperature trends in the lower stratosphere within the tropics and mid-latitudes. One conclusion is that most CCMVal2 models (in particular the multi-model mean) cannot reproduce observed trends in ozone from 1998-2017, while being able to capture temperature trends over the same period. They argue that this is only possible due to offsetting biases in simultaneous modeled SVW trends. As another important point, the authors argue that some models are better at capturing mid-latitude ozone trends inferred from observations than others, which in turn appears to be related to how lower stratospheric isentropic mixing is modeled in each case.

[Figure]

Without a doubt, the authors address an interesting but also highly complex topic. Their analysis therefore also requires particular care and has to be put into the context of the vast associated uncertainties. This makes it very difficult to study ozone and other trends over such short periods of time, which, in turn, links back to some weaknesses in their methodology and datasets used, which need either to be addressed or at least clearly highlighted to raise more awareness around them. Some of these challenges are already discussed in the paper, especially towards the end. However, currently, these uncertainties are not sufficiently reflected in the abstract, for example.

Major comments:

- The most concerning aspect are potential robustness issues: the authors consider very small trends that may or may not be due to actual climate change/MPA trends or simply artefacts of internal variability. On top of that, the observations are subject to uncertainties and the trends are also calculated differently (and the data preprocessed) for observations and models. At least this is how I understand section 2.2. The method to calculate the trends is also approximative. Overall, this implies that the main results might well arise from complex error propagation that for me as a reviewer is difficult to see through. This does not mean that the results may not be interesting or worthy of being published as a point of discussion. However, I also feel that some statements in the paper would ideally be tuned down and these uncertainties reflected appropriately and discussed more extensively. In particular, given the uncertainties and different methods to estimate internal variability contributions for models and observations, I have doubts about how well the 'accelerations' (second-order derivatives) in Figure 1c/d can actually be compared and how robust such a comparison can be.

- Is the singular attribution to SVW not too simplistic? Could dynamical heating not also play a role? Dynamical heating can also be quite different from what happens in the real world. In general, I consider ozone, temperature and dynamical trends a coupled problem, where cause and effect are difficult to distinguish. Would lower SVW trends not also be strongly influenced by model differences in isentropic mixing for example? How about differences in radiative transfer codes?

- The use of CCMs with SSTs different from the ones from observations makes me doubt if we can at all expect the models to perform similar to observations over this short time period. If, as a result, the DLM analysis is carried out differently, can we at all expect the same results (which will depend on these aspects of variability)? From the current text, this is at least not sufficiently justified. Do all CCMs actually use different /the same SST fields? Would we expect models (or subsets of them) to be consistent in terms of SST variability, which is surely connected to lower stratospheric ozone variability due to well-known effects of the ENSO etc?

- In the same vein, the use of a single radiative transfer model for the FDH calculations is necessarily imperfect, as different radiative transfer schemes themselves will contribute to the temperature trend differences among models.

- SVW trends can be very different for CCMs (see your own Supplementary figures). Re your trends in Figure 1b: how would the same trend for SWV look like of you took the model median, or plotted trends for all individual models? Would you still come to the same conclusions?

- How does the effective vertical range for lower stratospheric ozone trends compare for models and observations? How is lower stratospheric ozone defined in terms of the vertical range covered? Could different vertical dataset resolutions play a role in the differences you find?

- Re all trends you show: given that the models have different SST fields: would we really expect the MMM to be able to reproduce historic trends? Would we not

better ask if any of the ensemble members in the multiple CCM runs can reproduce the observed pattern of lower stratospheric ozone decreases? You might argue that models with multiple ensemble members might be consistently offset from observations (your Supplementary). However, did those different ensemble members actually use substantially different SST fields? Could a lack of skill in modeling SST variability in the first place be responsible for the apparent inability of models to capture lower stratospheric trends, i.e the biases are introduced somewhere else in the system unrelated to chemistry and stratospheric dynamics? If one ensemble member can reproduce historic trends, is it then in the realm of possibilities in the modeling world to reproduce observed trends, so to speak? If yes, can you still come to such strong conclusions concerning the models' skill to reproduce past trends?

Other comments:

- l.9: 'an increase'

- l.96-109: see above. Do we expect the MMM to be able to reproduce such a short period of time on average, or are we looking for individual ensemble members for this?

- l.138: typo?

- l.179: any particular reason why you used those 12?

- Figure 1: why are units/scalings for (c) and (d) not consistent?

- l.235-255: I am still not convinced that the MMM should agree with the single-member observations. Do you have any theoretical justification for that?

- l.283: typo

---

## Editor Comment (EC1) · Farahnaz Khosrawi (Editor) · 1 Jul 2020

First of all, I would like to thank both referees for the time and effort they have spent in reviewing the manuscript by Ball et al. Referee 3 raised a lot of criticism and rejected the version of the manuscript published in the discussions. However, although some of the criticism is justified (and has of course been taken into account by the authors in the revised version of the manuscript) I had the strong feeling that there is a conflict of interest. Therefore, after critically reading the manuscript myself I came to the conclusion that the manuscript will be suitable for publication in ACP after revision based on the referee comments and therefore encouraged the authors to do their revision.

[Figure]

This has been done now and the authors have accordingly considered all comments and criticism given by the referees and significantly improved their manuscript. Therefore, after critically reading the revised manuscript I came to the decision to accept this manuscript without a second round of reviews, but after consideration of some technical corrections.

---

## Author Response (AR1)

**Response to reviewers regarding**
**"*Inconsistencies between chemistry climate model and observed lower stratospheric ozone trends since 1998*"**
**by William T. Ball et al.**

**General**

We wish to thank the referees for taking the time to review our manuscript and provide perspective and input on how to improve it. As such, we have made steps to emphasize and improve the clarity of the message, and the meaning of the results.

We emphasize that the goal of this study is to perform (at this stage) the first direct comparison of ozone changes in the lower stratosphere between observations and free-running chemistry climate models; previously this had only been done with nudged (specified dynamics) and chemistry transport models. Given that observed and free-running modelled mid-latitude lower stratospheric ozone trends do not generally agree, it was important to assess if other physically related quantities exhibit the expected behaviour given the ozone trends (i.e. temperature), and provide a way to resolve the physical inconsistencies (i.e. by considering SWV). Many questions still remain, most notably as to why ozone trends do not agree. Nevertheless, while we agree in particular that comparisons have been performed between various ozone, temperature and stratospheric water vapour metrics within the CCMVal-2 project, the specific comparison we perform here (region, timescale) has not be done before and the results presented in CCMVal-2 cannot be directly used to understand what we show here.

Another key point, raised by both referees, relates to us not considering CCMI data. We agree it is overall beneficial to include as many up-to-date models as possible. To that end, we are aware of a parallel investigation by another team that has many similarities to our approach here, but using CCMI data (one of the coauthors here is involved in that study). As such, our perspective is that these two studies will complement each other.

Please note that we have followed advice put to us by referees and editor and added 'ozone' to the title. Additionally, Andrea Stenke is now involved and an author due to her expertise and knowledge of chemistry climate modelling, as a model developer and from her involvement in previous model intercomparison projects.

In the following we present the referee's comments (black) and our associated responses (blue).

**Anonymous Referee #3**

The motivation for this study is as follows: While CCMs show tropical O3 decline over the past 20 years that is likely driven by increases in tropical upwelling, models do not produce the (observed) decline in midlatitude O3; rather, they show an increase. The authors use a fixed dynamical heating model to estimate the impact of the negative O3 trend on the temperature

trend. The result is that the observed temperature trend is consistent with the observed O3 trend.The conundrum the authors find is that in spite of the disagreement between model and obs midlatitude O3 trends, the models and obs get similar temperature trends – this is not the physically expected response. The authors propose that the explanation for this is that the models have stratospheric water vapor trends that are opposite to those observed, thus creating a dynamical heating term that opposes that of the simulated O3 trend, leading to fortuitous agreement with the observed temperature trends. A sort of 'two wrongs make a right'. This lack of agreement between the CCMVal-2 models and observations motivates this study, whose intent is to explain what's wrong with the models and thereby help improve them and increase confidence their O3 projections.

It is important that we be clear on the intent of our work. First, a main motivation is to perform the first direct comparison of *free-running* CCMs updated to 2016 because, prior to this, no direct comparison for a similar period had been performed using the same analysis approach.

While REF-B2 is not the optimal version of the CCM data to do the comparison with, as it is used for long-term future projections without consistently including external and/or sea surface temperature and ice forcings, we believe it is appropriate as this category of data allows for a comparison up to 2016. Neither CCMVal2 REF-B1/B2 nor CCMI REF-C1/C2 have historical forcings/boundaries that go up to 2016. REF-B2 is used for future changes in the ozone layer for the 2014 ozone assessments, and the estimated changes are very similar to the 2018 report based on CCMI REF-C2 We now include some discussion on this in the text in section 2.2.

A question that naturally follows, as was raised by the referee (below), is perhaps there is an issue with the observational data. So, it is important to identify in other quantities that should respond to ozone if the signal can be seen there too. Thus, once we performed the comparison between observations and CCMs, and showed them to differ, a key question was to follow up the statement by Maycock et al., 2018 (GRL) that "*The models and an extended satellite data [...] show weaker global stratospheric cooling over 1998–2016 compared to the period of intensive ozone depletion (1979–1997). This is due to the reduction in ozone-induced cooling from the slowdown of ozone trends and the onset of ozone recovery since the late 1990s.*" In other words, *if the observed negative ozone trends are real* and now diverge from the models, then it should show up in the difference between observed and modelled temperature trends. However, as Maycock et al., 2018 showed with CCMI models, and we show here with CCMVal-2 models, that is not the case: temperature is in agreement. This is precisely why we then looked at SWV, which we show is sufficient to first order (given the opposite tendencies) to explain the difference.

It appears to us, and to those we have discussed this with within the wider community, that this is not well understood or widely known - again, this specific comparison has not been made or presented within the CCMVal-2 report. Therefore, the results we have found are worth communicating to the wider community, but ascertaining the reasons why is beyond the scope of the publication and requires engagement across the community.

Nevertheless, we accept that perhaps this narrative is not communicated as well as it could be, so we have made the following changes to the abstract (bold): "On the other hand, mid-latitude lower stratospheric ozone is observed to decrease, while CCMs **that specify real-world historical meteorological fields show instead an increase up to present day. However, these cannot be used to simulate future changes; we demonstrate here that free-running CCMs used for projections also show increases**."

Much of the knowledge regarding model problems that is so 'urgently needed' is already available in the SPARC (2010) CCMVal evaluation. It has over 400 pages of detailed analyses showing why these models don't match observed O3, temperature, trace gases, variability and more. Chapter 2 is very useful because it details how each model differs in its representation of radiative, chemistry, and dynamical processes, boundary conditions, etc. Chapter 3 is all about radiative processes and the information presented reveals much about how these models can be expected to respond to changes in radiatively active trace gases, i.e., O3. Chapter 5 reports on the transport issues affecting the credibility of their ozone simulations, in particular how well they represent tropical ascent and mixing out of the subtropics – topics so very relevant to how models' circulations will be respond to increasing GHGs and alter future ozone distributions. Chapter 8 examines simulated O3 variability and whether models have the necessary processes to simulate that variability (spoiler alert: they don't). This report may be 9 years old but there is much about models and the physical processes requisite for simulating ozone that must be understood before undertaking an investigation of why CCMs don't do a particular thing right.

The reviewer is correct that there is a wealth of information in the CCMVal-2 report. However, we respectfully disagree that there is *sufficient information* in that report (see below) to determine the exact cause (i.e. which model parameters, resolution, assumptions etc) behind the divergence between models. Our paper also does not solve this and this is not the goal. However, it is the goal to do a comparison and highlight the long-term differences between the models and observations. The vast majority of the report does a painstaking, but necessary, comparison of many metrics to assess multiple model attributes. This is typically done with a 'one at a time' comparison, and it is difficult to tease apart the confounding, conflicting, and reinforcing effects of each of these. Often, due to the number of models being discussed, and due to the sheer difficulty in separating contributions from different effects (chemistry, radiation, dynamics, parameterisations etc), the authors were constrained to make suggestive comments.

Further, we admit there is some ambiguity in the final statement of the abstract related to understanding the models, and so we have amended it to read: "***The reason CCMs do not exhibit the observed changes needs to be identified to allow models to be improved in order to build confidence in future projections of the ozone layer.***"; we removed 'urgently' from the abstract.

We now address sub-points of the above paragraph from the reviewer. Here, we are not criticising the report, on the contrary we extract the relevant information for our study and indicate where these have been added in the text. Nevertheless, we felt it important to elucidate

with evidence why we disagree, and why our comparisons are only as a first step to identifying and fixing problems.

Chapter 2 is very useful because it details how each model differs in its representation of radiative, chemistry, and dynamical processes, boundary conditions, etc. Chapter 3 is all about radiative processes and the information presented reveals much about how these models can be expected to respond to changes in radiatively active trace gases, i.e., O3.

We agree there is much information available here. However the *"inferences in this section are suggestive" (3.3.1)* with multiple possibilities discussed (e.g. clouds and aerosol treatment, relaxation time and transport, heating rates, biases in $CO_2$/SWV/$O_3$ as controllers of radiative energy (3.4) - although these biases are not thought to affect long-term trends - etc). Even these topics need further consideration. We do not cover these topics either. But, the added value of our study is that we have extended the analysis and done direct trend analyses with respect to the CCMVal-2 report; these CCMVal-2 trend comparisons have not been performed in this manner elsewhere and it raises questions not previously considered, in the light of the ozone **observations**. We have added the following to the introduction: **"As such, while the CCMVal-2 report provides an extensive comparison of the models with observations, across multiple timescales and metrics (including, e.g., transport, heating rates, radiative transfer codes, and boundary conditions; see chapter 3 of CCMVal-2 report), ozone trends over the 1985-2016 period were not. Here we consider the specific issue of recent ozone trends."**

Chapter 5 reports on the transport issues affecting the credibility of their ozone simulations, in particular how well they represent tropical ascent and mixing out of the subtropics – topics so very relevant to how models' circulations will be respond to increasing GHGs and alter future ozone distributions.

Chapter 5 does discuss transport, but the overall point is that models have significant problems with respect to transport on multiple scales and that (ch 5.4) circulation in the models *"continue to be an important area of research"*. These transport issues are, of course, relevant to ozone, particularly in the lower stratospheric region we focus on (ch 5.3.1.4): "*Overall, models [...] do not perform as well at 100 hPa [...]. The seasonal cycles are worst in the SH at 100 hPa [... and ...] suggests that the strength of the circulation is too weak in the lowest levels of the SH stratosphere*." Our study adds value by focusing particularly on trends and a time-period not considered in the report, and again highlights the need to point out that a discrepancy between observations and modelled ozone trends persists especially given that the report's grades on transport diagnostics (ch 5) *"suggests that transport deficiencies exist in most models"*. As we indicate in our study this is likely related to transport - but the CCMVal2 report only makes an implicit connection to ozone and highlights that much work is needed still.

We added to the penultimate paragraph of the discussion section: "**For example, the CCMVal-2 report provides an extensive intercomparison and discussion of the deficiencies across CCMs in simulating transport (chapter 5), particularly at around 100 hPa in the lower**

**stratosphere, and the modelling of the QBO was considered 'too primitive' to make an assessment at that time (chapter 8) and is an issue needing further work, especially with respect to its impact on modelled ozone. While focus in these examples was on variability, it is not surprising that trends may consequently differ too. As such,** untangling and identifying the aspects responsible for the spread in trends remains an important focus in model evaluation."

Chapter 8 examines simulated O3 variability and whether models have the necessary processes to simulate that variability (spoiler alert: they don't).

Chapter 8 focuses on natural variability, and the agents that induce it, e.g. seasonal, solar irradiance, QBO, ENSO and volcanic variations (wave parameterisations also play a role). It points out the dominant influence of transport on lower stratospheric ozone, and that the aforementioned agents are important to transport in this region. The reviewer is correct that in many cases the models do not have the right, or even any, perturbation or forcing included (e.g. five models in the REF-B1 case do not have a solar forcing). This is particularly a serious issue for the QBO (as pointed out by the reviewer below), where only 8 of the 18 models (Table 8.4 of the CCMVal-2 report) contain realistic variability from nudging, and the others perform poorly in terms of the magnitude of variability even when some QBO-like variability is produced. We entirely agree with the reviewer's point that deficiencies in the QBO *may be* an important reason the models do not reproduce the trends, *but this is not yet proven.* Further, the report does not fully evaluate the QBO and goes on to state in section 8.9 that the *"QBO signal in ozone could be a second candidate for a quantitative evaluation. However, the modelling of this phenomenon in CCMs is in a too primitive stage to apply performance metrics."* and that *"QBO modelling in the CCMs as implemented for CCMVal-2 therefore remains an outstanding problem"*. We added a short comment on this (see previous comment, above). While beyond the scope of this work, two of the coauthors are preparing an in depth analysis of this specific aspect of the models (see Stenke et al., EGU 2020, https://doi.org/10.5194/egusphere-egu2020-16682).

The authors justify their use of the CCMVal2 simulations rather than the CCMI runs because these models (in some ways?) are 'still representative of the state-of-the-art'. I think it is a mistake not to use the CCMI runs because although some models may be the same as in CCMVal2, others have made improvements. (Shouldn't the goal be to improve the current state of modeling?) Whether you use CCMVal2 or CCMI simulations, the conclusions of the SPARC (2010) report still provide a relevant starting point for a study like this.

We agree with the sentiments of the reviewer, although our understanding from the literature is that CCMVal2 and CCMI models are overall similar in their simulation of ozone changes (see Discussion section in the manuscript). Indeed, no comprehensive report for CCMI, as for CCMVal-2, exists. We have been collaborating with another research team to perform a similar analysis using CCMI data, so this will also be published in the near future as well and these will therefore be complementary studies. Some of the authors here are also preparing a manuscript on a related topic with the preliminary analysis showing little difference in the long term ozone

trends between CCMVal-2 and CCMI, except in most cases the CCMI models cluster more closely around a similar MMM.

Another key point is that the CCMVal-2 models (and specifically REF-B2) were used in the WMO ozone assessment report 2014 in comparison with the observations. Therefore, they form a very relevant comparison data set for the study here, and little has changed with respect to the WMO ozone report for 2018 that used CCMI data (although not presented in the same way) from the REF-C2 simulations, i.e. equivalent to the CCMVal-2 REF-B2 simulations that do not have the observed forcings.

Using these CCMs' transport behavior to support the interpretation of observed O3 trends in terms of ascent and mixing is problematic given the many model transport problems diagnosed in Ch. 5 of the SPARC report. See in particular Figure 5.20 that evaluates tropical ascent and meridional mixing. Most of the models used in this paper did rather poorly here. When a model does a poor job at representing a physical process, it should never be assumed that the simulated process will (magically) respond in a physically meaningful way to changes in forcing (e.g., increasing GHGs).

We did not previously consider the transport diagnostics from the CCMVal-2 models, though did refer to the CCMVal-2 report. Now, we have calculated upwelling changes from the CCMVal-2 models, where available, and added these to the reanalysis results in Fig. 4; the diagnostics for effective mixing calculations were not available; model results are largely in agreement with w*, although the model mean is lower than the reanalysis at 100 hPa.

An assessment of the problems in transport, which is considered in a lot of detail in the CCMVal-2 report (although no report exists for CCMI), is *not the goal of this paper*. Our goal is to elucidate that a divergence in ozone trends is apparent , understand if (independent evidence from) temperature is consistent with this, and to put emphasis on a need to understand and diagnose what is going on. We believe this is valuable in its own right, even if it doesn't resolve the issues.

We have added the following text to the Discussion section to point out that deficiencies in transport are well known from the report: **"Nevertheless, large-scale CCM transport deficiencies exist in most models, such that while there is consistency across models, comparisons across multiple metrics indicate shortcomings in transport, e.g. even in the representation of seasonal cycle variability in southern hemisphere lower stratosphere transport (CCMVal-2 report)."**

Furthermore, given that these CCMs have many different radiative, chemistry, and dynamical problems, using the multi-model mean (MMM) is a bad idea. The MMM is a mash-up of correctly and incorrectly simulated (or missing) processes. This manuscript works from the assumption that something physically meaningful can be derived from their analysis of the

MMM. That's not possible. As an example, here are some relevant conclusions from the Ch. 3 summary of the models' radiation evaluation that speak to what you get with a MMM:

". . .5 out of 18 CCMs show biases in their [temperature] climatology that likely indicate problems with their radiative transfer codes." "Problems remain simulating radiative forcing for stratospheric water vapour and ozone changes with a range of errors between 3% and 200% compared to LBL [line by line] models." "The stratospheric water vapour forcing has errors of over 100% between the models (Figure 3.12)."

We accept the MMM is not physical. But we do think it is representative of an aggregate of model performance, it is relevant in many studies, and is a key part of the ozone assessments. As such, much research is motivated by these MMM results, and so it is an important quantity to focus on. Further, we provide the breakdown across models precisely because we want readers to get an indication of what the MMM represents and what the spread across models is. We have alluded to this point in our other additions to the text.

Did you know water vapour in some of the models can't respond to climate change because a water vapour climatology was used?

Yes, we did know, and we explicitly stated (and now added UMUKCA-METO) in the manuscript that *"[...] we performed another sensitivity test to see how removal of several CCMs [...] chosen due to [...]: CAM3.5 (missing results in the upper stratosphere), UMUKCA-UCAM (climatological SWV), UMSLIMCAT (no SWV available), and CNRM-ACM (no SWV available); again the results remained similar, so we do not remove them for the full analysis performed in the paper."* and *"UMUKCA-UCAM SWV is climatological and displays no change."*

We did not evaluate UMUKCA-METO as it was not available at the time - but similarly has a climatological SWV - it has subsequently become accessible again and we have integrated it into the manuscript. UMSLIMCAT and CNRM-ACM were similarly unavailable for SWV, but are now also available and have also been included. The above quoted section has been rewritten as a result of these changes.

I also read in the report of a situation where the MMM quantity (I don't remember which one) actually got a higher grade than the models did individually on a particular evaluation.

That is correct; this was in section 3.6: *"The multi-model mean has a higher grade than all but one model (WACCM) which indicates the value of multi-model studies"*.

Bottom line: the MMM is not a quantity from which you can derive physical meaning.

We agree with the reviewer, and this is why we have generally provided estimates from the individual models for each of the metrics as well throughout and again refer the reviewer back to similar points we have already made in the manuscript (lines 96-100): *"From a modelling perspective, averaging multiple CCMs into a MMM suppresses unforced natural variability and therefore reduces uncertainties in trend analyses; it can also lead to a loss of information*

*regarding the sensitivity of CCMs to a changing state, and the range of responses to drivers; warnings against such averaging to understand CCM efficacy have been raised before (Douglass et al., 2012, 2014)."*

Later (lines 413-416) we stated that *"The mechanism proposed here -- with SWV and ozone driving the majority of temperature changes -- does not fully explain the different changes in temperature between each CCM (Fig. 2); this will require a deeper, case by case examination of how each model is operating."* In this point, the aim was to be upfront that each model has its own response that doesn't agree exactly with the MMM (and can be seen by examining the figures, which we refer to in the text). To address this, we have preceded the second sentence with the following: ***"We point out that a MMM does not necessarily provide physically meaningful insights (CCMVal2 report) and may provide confidence in CCMs that show similar, e.g., trends for different reasons. That said, a MMM does provide an aggregate metric for the general behaviour for a group of CCMs when individual CCMs are not downgraded or removed for their poor performance, with the assumption that the influence of poor physical representation is diminished through the act of averaging."***

The second sentence in our amendment is supported by the CCMVal2 report (among other journal articles), an example of which comes from section 5.2.2.4: "*The MMM tropical mean age profile closely matches the profile of this cluster of models with good performance; the models with poor performance largely cancel each other out.*" Other examples exist in the report.

In conclusion, while we agree with the reviewer about the physical meaning of a MMM, the MMM is a standard by which information is aggregated in modelling and intercomparison studies, e.g. especially the WMO ozone assessment reports, as we mentioned above. Therefore, it is important that we consider it to allow for comparisons and as a metric to combine information across models. Given that our study does not intend to solve all CCM problems, it is reasonable to summarise the efficacy of the models to simulate trace gas and temperature behaviour through the MMM.

I'm curious why the authors excluded 6 of the 18 CCMs models for the MMM. The excluded models (AMTRAC, EMAC, E39, GEOSCCM, UMetrac, and UMUKCA-Meto) all provided O3, H2O, and T outputs.

AMTRAC, EMAC, E39, UMetrac were not available on the BADC dataserver at the time of writing and still are not (the REF B1 runs also stopped in 2005, so cannot be used). As mentioned earlier, UMUKCA-METO was also not available, but subsequently is and we have now included it. GEOSCCM REF-B2 is only available for 2000-2100, so was not valid for our analysis.

On a different topic, the authors report that lower stratospheric ozone has negative trends, a conclusion not broadly agreed upon by the community. See for example Steinbrecht et al.

(2018), WMO (2019), and the LOTUS report (2019). None of these publications finds statistically significant O3 trends below the upper stratosphere.

Agreed. The aforementioned publications do not show significant ozone decreases. However, Steinbrecht et al., 2017 (not 2018) states *"there might be an indication for decreasing ozone in the tropics and at northern midlatitudes",* although they go on to express concerns due to instrumental issues. In many of the latitude-pressure spatial results from ozone composites in Steinbrecht et al 2017 and the LOTUS report (which the WMO conclusions are based on, so these are not independent), there are regions of non-significant decreases; issues with, e.g. SAGE-MIPAS-OMPS, which show an increase in the lower stratosphere (even in the tropics), are discussed in the LOTUS report. On top of that, there are publications that support the finding of negative trends in the mid-latitude lower stratosphere, and these are already referenced in the manuscript. The series of papers by Ball et al., (2017, 2018, 2019) have made efforts to address some of the instrumental issues, and to investigate the sensitivity of trend estimates to unaccounted for natural variability. These are by no means exhaustive, but they remain the most up-to-date attempts to consider these issues.

An important point, we would argue, is that whether or not the trends are negative or flat (but, rather, evidence lacks for an increase at mid-latitudes), the CCMs are reporting an increase over the period from 1998 to 2016. This raises concerns about their representativeness of the true atmosphere under increasing GHGs (which the reviewer has laid out above also) and needs first to be demonstrated (e.g. the aim of this paper) and then tackled. If it is natural variability, then models do not appear to be reproducing this (see earlier comments of dynamics in the CCMVal-2 report) and we may be overconfident in an increase of ozone. But, as the reviewer quite rightly points out, there are multiple (and different) problems in CCMs that may mean they do not capture dynamical variability, or those induced by long term increases in GHGs, such that large scale long-term changes are not being adequately reproduced. We argue that we demonstrate these differences in this paper, and believe this to be the most important, though quite straightforward, result of the study.

To address this point specifically, we have added the following (bold text) to the introduction where there is discussion of the downward trends: *"Recent findings indicate that contrary to chemistry climate model (CCM) predictions, ozone in the lower stratosphere **does not yet display positive changes since the turn of the century (Steinbrecht et al., 2017; LOTUS 2018), and indeed there is evidence that it may have** continued to decrease over 1998-2016..."*

Steinbrecht et al. note that the search for ozone trends is complicated by ozone variations not caused by declining ODSs, such as variability in or changes to the BrewerDobson circulation. The role of circulation variability on ozone trends is acknowledged by the authors, yet they seem not to recognize that the CCMs' inabilities to provide anything close to observed interannual variability in stratospheric composition is one of the greatest shortcomings that interferes with the ability to simulate credible O3 projections.

We agree - the point of our paper is to highlight that if (i) there is disagreement in the projections to present day, and (ii) the models are still being used to make projections, then it is appropriate to do a comparison and point out these issues (whatever the reason for that may be, and dynamics is certainly a key potential culprit, as pointed out by the reviewer, but also multiple publications, including Ball et al., 2019). We do not agree, however, that we can extrapolate the poor representation of interannual variability to conclude that multi-decadal projections are wrong - and we are careful not to make that assumption - but we do think that may be a possibility and it needs to be determined if that is indeed the case.

The QBO is the greatest source of stratospheric composition variability and these models either have no QBO or an unrealistic one. This is the elephant in the room with respect to 'why don't the CCMs get a negative or flat O3 trend over the past 20 years'. See Chapters 4, 5, and 8 in the SPARC report.

This is certainly a potential candidate, though there is as yet no clear evidence that the poorly represented QBO *is the cause* of not reproducing the trends. A separate publication is in preparation focused specifically on modelled QBO lower stratosphere transport, where the SOCOL model is specifically focused on and that having a realistic QBO (though nudged) does not solve the problem (see Stenke et al., EGU 2020, https://doi.org/10.5194/egusphere-egu2020-16682). The CCMVal-2 report *does not* demonstrate this to be a cause for the differing lower stratospheric trends since trends in this region were not a focus in the report. Further, evaluation of the QBO in the CCMVal-2 report was limited (section 8.9 of the report) and there was no consideration of its impact on the lower stratosphere; other publications have focused on variability rather than trends (see, e.g., Punge and Giorgetta, ACP, 2008). We note from our ongoing work that although QBO-interannual variability is not well reproduced in the lower stratosphere mid-latitude region in free running models, it is also not the cause of the divergence of models and observations, since the fully nudged models reproduce the QBO induced variability, but not the trends. Understanding and progress is certainly needed here (see earlier text addition, above).

The overarching motivation of this paper – to improve the credibility of CCMs– is fine. The analysis of individual model behavior, especially in a model with vetted radiative, chemical, and transport processes, may produce useful insights into CCM needs. However, the problem stated by the paper's title – inconsistencies between chemistry climate model [sic] and observed lower stratospheric trends – cannot be solved with the chosen approach. The results presented rely on analyses of the physically meaningless MMM, which is fundamentally not a valid approach; for this reason I do not recommend publication.

We agree, again, that we cannot solve the problems with CCMs with respect to the observations; again, that is not our goal. In summary, the goal of the paper is to highlight the inconsistencies and demonstrate the need to resolve them given the importance of the ozone layer to protect the biosphere. The MMM is a useful metric, though not necessarily insightful to the physical reasons, but given its widespread use in making estimates for future recovery (Dhomse et al., 2018; WMO 2018) or for assessing models (CCMVal-2 report), and for policy

decisions and public discourse, we think it is useful to highlight the MMM results in a succinct way, while providing individual CCM results so that the underlying spread is revealed. We leave many open questions that we hope the community, and in some cases ourselves, will follow up in the future.

Anonymous Referee #1

The paper 'Inconsistencies between chemistry climate model and observed lower stratospheric trends since 1998' by Ball et al. discusses recent ozone, stratospheric water vapor (SVW) and temperature trends in the lower stratosphere within the tropics and mid-latitudes. One conclusion is that most CCMVal2 models (in particular the multi-model mean) cannot reproduce observed trends in ozone from 1998-2017, while being able to capture temperature trends over the same period. They argue that this is only possible due to offsetting biases in simultaneous modeled SVW trends. As another important point, the authors argue that some models are better at capturing mid-latitude ozone trends inferred from observations than others, which in turn appears to be related to how lower stratospheric isentropic mixing is modeled in each case.

Without a doubt, the authors address an interesting but also highly complex topic. Their analysis therefore also requires particular care and has to be put into the context of the vast associated uncertainties. This makes it very difficult to study ozone and other trends over such short periods of time, which, in turn, links back to some weaknesses in their methodology and datasets used, which need either to be addressed or at least clearly highlighted to raise more awareness around them. Some of these challenges are already discussed in the paper, especially towards the end. However, currently, these uncertainties are not sufficiently reflected in the abstract, for example.

We accept the points raised here and are more than happy to elucidate additional uncertainties. For example, we have added the following (bold) to the last part of the abstract:

"Together, our results suggest that large scale circulation changes expected in the future from increased greenhouse gases (GHGs) may now already be underway, but that most CCMs are not simulating well mid-latitude ozone layer changes. **However, it is important to emphasize that the periods considered here are short and internal variability that is both intrinsic to each CCM and different to observed historical variability is not well characterised and can influence trend estimates. Nevertheless,** the reason CCMs do not exhibit the observed changes  needs to be  **identified and improved to build** confidence in future projections of the ozone layer."

Major comments:

The most concerning aspect are potential robustness issues: the authors consider very small trends that may or may not be due to actual climate change/MPA trends or simply artefacts of

internal variability. On top of that, the observations are subject to uncertainties and the trends are also calculated differently (and the data preprocessed) for observations and models. At least this is how I understand section 2.2. The method to calculate the trends is also approximative. Overall, this implies that the main results might well arise from complex error propagation that for me as a reviewer is difficult to see through. This does not mean that the results may not be interesting or worthy of being published as a point of discussion. However, I also feel that some statements in the paper would ideally be tuned down and these uncertainties reflected appropriately and discussed more extensively. In particular, given the uncertainties and different methods to estimate internal variability contributions for models and observations, I have doubts about how well the 'accelerations' (second-order derivatives) in Figure 1c/d can actually be compared and how robust such a comparison can be.

The acceleration component is simply there to emphasize the covarying nature of physically dependent variables in either the models or in the observations, though not between models and observations. Nevertheless, we have now added supplementary plots that allow for a direct comparison of the no-regressor observational trend estimates with the regressor-included observational estimates (now Fig S2 in the supplementary materials). In other words, we have applied the no-regressor analysis to the observations too (the opposite is not feasible due to missing information and the need for model-dependent regressors). What we find is that while uncertainties usually increase, the mean changes are typically similar to those of the observations when using the regressors. This can mean one of two things: that the estimates are orthogonal to the regressors and subsequently the regressors do not bias the trend component, or that the regressors do not provide much attribution and therefore play little part in influencing the trend estimate. Either way, this then implies that the comparison between observations and CCMs is reasonable even with different approaches to estimating the trends, and further that the uncertainties are likely to decrease if regressors were available to use. Of course, this assumes that models and observations behave in the same way, but if this is not true, then the situation is more serious.

In addition to Fig S2 showing the sensitivity of observations to use of regressors, or not, we include a brief mention in section 2.3 (DLM) and section 2.2 (CCMVal-2 models) as follows: **"We performed a sensitivity test on the observations by applying DLM with and without regressors (Fig. S2) to test the impact on the trend. We found that the trend estimate does not change much between the two cases, although the uncertainties usually increase when no regressors are used."**

Is the singular attribution to SVW not too simplistic? Could dynamical heating not also play a role? Dynamical heating can also be quite different from what happens in the real world. In general, I consider ozone, temperature and dynamical trends a coupled problem, where cause and effect are difficult to distinguish. Would lower SVW trends not also be strongly influenced by model differences in isentropic mixing for example? How about differences in radiative transfer codes?

The reviewer is right in pointing out that these variables (ozone,SWV and temperature) are intrinsically related to each other. That said, the radiative signature of both ozone and water vapor in the stratosphere is sizable and can be clearly detected (Clough and Iacono JGR , 1995). As a result, both ozone and SWV have a large impact on the stratospheric radiative equilibrium and thus in temperature, as discussed in the literature (see the CCMVal-2 report and Shine et al., 2003). However, as correctly pointed out by the referee, the dynamics, via e.g. changes in upwelling, can drive large departures from radiative equilibrium and thus also change temperatures. In our paper, we isolate the impact of radiative processes via the PORT calculations, which provide the stratospheric temperature adjustment driven by changes in radiative heating rates, assuming the dynamical heating (i.e. adiabatic warming/cooling) remains unchanged; this is the FDH approximation (Fels et al., 1980). By contrasting the temperature adjustment ("$T_{fdh}$") imposing changes in SWV and ozone against the temperature change derived from observations and model simulations ("$T_{free}$"), one can assess the importance of radiative processes related to SWV and ozone in driving the observed/modeled temperature trends. We show in Fig. 5 that "$T_{fdh}$" closely resembles "$T_{free}$", which is a strong indication that ozone and SWV drive a substantial portion of the (albeit small) lower stratospheric temperature trend in models and observations. We also tested the impact of dynamical heating, by calculating the temperature change driven by upwelling ("$T_{dyn}$"), but found that this term provides a worse match with $T_{free}$ (not shown). Hence, we believe that this is sufficient evidence supporting our claim for a "causal" effect of SWV and ozone on lower stratospheric temperatures.

The reviewer also raises a point about the separability of the local temperature effects of SWV and ozone. We tested the "linearity" of the individual contributions of SWV and ozone and found that they are linearly additive to within 3-5%, in line with Forster et al., 1997. Finally, differences in radiative transfer codes indeed may play a role in the inter-model spread, but we think the impact of this source of uncertainty will be small, given the good match between our FDH estimate ("$T_{adj}$") computed from one single model, and the trends obtained from 'free-running' experiments with different models.

The use of CCMs with SSTs different from the ones from observations makes me doubt if we can at all expect the models to perform similar to observations over this short time period. If, as a result, the DLM analysis is carried out differently, can we at all expect the same results (which will depend on these aspects of variability)? From the current text, this is at least not sufficiently justified. Do all CCMs actually use different /the same SST fields? Would we expect models (or subsets of them) to be consistent in terms of SST variability, which is surely connected to lower stratospheric ozone variability due to well-known effects of the ENSO etc?

The reviewer raises a very important issue that is currently not addressed in the paper. The reviewer is correct that ENSO should affect variability in the lower stratosphere. In terms of attribution through linear regression in the observations, the ENSO component explains only a small fraction, and using or excluding regressors has little effect on the trend estimate (see earlier). REF-B1 does essentially have only one SST (HadISST1; except MRI that uses a hybrid

of HadISST1 and MRI-CGCM2.3.2). However, REF-B2 that is used here, considers a large range of SST boundary conditions:

- three use SST/SSI from **CCSM3**: *CAM3.5, ULAQ*, and *WACCM*;
- four use **HadGEM1**: *Niwa-SOCOL* (>2002 only), *UMSLIMCAT, UMUKCA-UCAM, and UMUKCA-METO;*
- all remaining models use their own, unique boundary condition SSTs/SICs, i.e. *CCSRNIES* (**MIROC** SSTs/SICs), *CMAM* (**Interactive** ocean), *CNRM-ACM* (**CNRM-CM3**), *LMDZrepro* (**OPA/LIM**), *MRI* (**MRI-CGCM2.3.2**), *SOCOL* (**ECHAM5-MPIOM**).

We have plotted the Nino 3.4 index from observations (https://climatedataguide.ucar.edu/climate-data/nino-sst-indices-nino-12-3-34-4-oni-and-tni) and those extracted from 1000 hPa temperature data for each model in Figs. 1 and 2 here:

[Figure]

Fig. 1. Nino 3.4 index from observations (thick, black in each plot) and sets of the models depending on if they use a specified SST for multiple models (panels 1, 2 and 6), the ensembles with the same model(s) (2, 3 and 5), or an interactive ocean (panel 4) as for MRI. All data have been normalised by subtracting the mean and dividing by the standard deviation of the data (NIWA is normalised post-2002 due to the discontinuity at the end of 2002 where SST

[Figure]

Fig. 2. Nino 3.4 index from models, coloured based on the same SST used (or single model-SST pairs in grey). Only the last 7 years are shown for clarity. See Fig. 1 for more information.

These plots, and knowledge of the underlying SST forcing helps answer the reviewer's question, that it is unlikely that ENSO is having a significant impact on the trend term, and that other factors are having a larger effect. From Fig. 2 of the manuscript and the plots provided above, the eight different ocean boundary conditions makes it apparent that:

- within-SST-groups (e.g. *CAM3.5, ULAQ*, and *WACCM)* there is a spread in responses (Fig. 2 in the manuscript) of similar a range to the overall spread between SST-groups (e.g. T at most lats, SWV in the NH, or O3 in the tropics);
- equally, and for completeness, the spread between unique boundary conditions is as large as those within the group of models with the same SSTs; neither this nor the previous statement is consistent across variables;
- SOCOL provides an example of the same model leading to a large spread in estimates (e.g., T in the SH, Fig 2g), which cannot be a result of the SST/SIC boundary conditions.
- the ensemble sets for each model show that there can indeed be a consistent response when using the same SST/SIC forcing across ensemble members, but it also shows that this is not sufficient, i.e. ULAQ shows large variations in the lower stratosphere between models, indicating that having a consistent SST/SIC does not mean you will reproduce the same response (this is not at all surprising, due to natural internal variability, but it reaffirms that SSTs might not be dominant).

Therefore, the use of multiple boundary conditions also makes it more difficult to disentangle what is driving each trend and how much can be attributed to SST/SIC changes, since the in-group spread and that of the independent (single ensemble) CCMs are similar.

Our overall conclusion is, therefore, that it is unlikely that SSTs/SICs are having a significant impact on the spread, or if it is, then its impact is limited to the combined range presented in Fig 2 (and SM figures). We have now added a discussion of SSTs/SICs as raised here in the discussion section with: "**One important aspect of the analysis performed here is that the CCMs do not include regressor terms, due the absence of information to make fair comparisons when using different sets of regressors, and since observations with and without regressors display similar mean trends (Fig. S2), this implies that the length of the timeseries is long enough to mitigate the effect of short-term behaviour from forcing agents such as ENSO on the trend estimates. Indeed, sea surface temperatures (SSTs) are expected to have a large impact on stratospheric variability, usually represented by ENSO variability. But our results appear to indicate that, at least for the model scenario (REF-B2) considered here, SSTs do not have a clear impact on the trend estimates, and it is likely that other factors have a more significant impact. For example, the range of trend estimates in SH lower stratospheric temperature between SOCOL ensemble members is as large as the range between all other models (Fig. 2g), despite using the same SST forcing (Fig. S8), while the set of other CCMs use seven other varieties of SST boundary conditions (see Fig. S8 and CCMVal-2 report). Similarly a large range of changes can be found between ULAQ, WACCM and CAM3.5 that all use the CCSM3 SST as a boundary condition. Counterexamples can also be found, but there is little consistency between the relative trend estimates of CCMs across variables (i.e. Fig. 2) depending on the SST boundary conditions.**

**So, the overall implication is that SST boundary conditions cannot be singled out as a major factor influencing the trend estimates, when other aspects of (atmospheric) internal variability or different CCM design appear to be responsible for a similar range or, more likely, larger impact on the stratospheric variability. For example, chapter 5 of the CCMVal-2 report provides an extensive intercomparison and discussion of the deficiencies across CCMs in simulating transport, particularly at around 100 hPa in the lower stratosphere. As such, untangling and identifying the aspects responsible for the spread in trends remains an important focus in model evaluation.**"

In the same vein, the use of a single radiative transfer model for the FDH calculations is necessarily imperfect, as different radiative transfer schemes themselves will contribute to the temperature trend differences among models.

The referee is right in that radiative transfer schemes are quite distinct in the different models, and differences in the parameterization of absorption bands and in the background state can lead to uncertainty in the radiative effect of ozone, SWV, carbon dioxide, etc. This may also lead to uncertainty in the calculation of the heating rates and thus in the FDH estimates ("$T_{adj}$"). However, we have reasons to believe that the uncertainty related to radiative transfer is going to

be rather small. In the Chapter 3 of the CCMVal-2 report (2010), Forster and colleagues did an extensive radiation code evaluation, by comparing LbL vs broad-band codes used in CCMs. In Table 3.7, the net heating rate near 70 hPa is estimated to be about 0.24 K/day in the LbL model. As shown in this same table, the CCMs deviate from this LbL value by at most 0.03 K/day - which means a 12% error. The net heating rate at this atmospheric level will be almost entirely determined by water vapor and ozone, so this result should apply to a good degree to the present paper. Hence, we expect that the error introduced by radiative transfer in the FDH estimates is on the order of 10-15%, and thus does not influence the main conclusions of our study.

SVW trends can be very different for CCMs (see your own Supplementary figures). Re your trends in Figure 1b: how would the same trend for SWV look like of you took the model median, or plotted trends for all individual models? Would you still come to the same conclusions?

The following plot, as requested by the reviewer (which goes with Fig S3c) shows that it is unlikely there would be much effect on the MMMs if the median was considered since, apart from Niwa-SOCOL (grey line, lowest in 2016), all the models cluster around similar trend values, distinct from the observations.

[Figure]

Fig. 3. DLM trend estimates for each individual model (see Fig 2 or Fig S1) for SWV at 83 hPa; this figure directly reflects the mean values for the distributions in Fig S1c; see that figure for colours linked to model names.

How does the effective vertical range for lower stratospheric ozone trends compare for models and observations? How is lower stratospheric ozone defined in terms of the vertical range covered? Could different vertical dataset resolutions play a role in the differences you find?

The altitude ranges were noted in the Fig 2 caption, but to clarify this we have added it to the main text at the beginning of the results section the pressure range for ozone partial column:

**"defined as 147-32 hPa in the mid-latitudes, 60-30, and 100-32 hPa in the `tropics',
30S-30N**". Certainly vertical resolution may be an important factor and is the reason why we use
147 hPa (historically chosen from observations) to avoid picking up the troposphere (the
tropopause varies seasonally but should remain below this level; on the longer term, Fig 7.37 of
the CCMVal-2 report shows the tropopause air pressure and for the model with the fastest
increase this is ~2 hPa decrease over 1998-2016, so will have little effect on the trends
assuming tropospheric ozone changes are also relatively small in the models). We have
performed, elsewhere (not shown), sensitivity tests by integrating from the tropopause (WMO
lapse rate definition) to 32 hPa, and we see negligible difference in ozone trends in those cases.
Model resolution can be around ~1 km in this region, so usually better than the satellite
observations that we use here. Even so, upper tropospheric ozone is thought to be increasing
but is approximately constant over the period considered here in the models, so this would
further reinforce our observational results here (tropospheric ozone in CCMI REF-C1 is
approximately constant (Revell et al., 2015) and ozone in the troposphere in Fig 7.37 implies
~0%/dec change in the tropical upper troposphere region, or increasing in the mid-lats, though
that was estimated over 1960 to 2100).

Re all trends you show: given that the models have different SST fields: would we really expect
the MMM to be able to reproduce historic trends? Would we not better ask if any of the
ensemble members in the multiple CCM runs can reproduce the observed pattern of lower
stratospheric ozone decreases? You might argue that models with multiple ensemble members
might be consistently offset from observations (your Supplementary). However, did those
different ensemble members actually use substantially different SST fields? Could a lack of skill
in modeling SST variability in the first place be responsible for the apparent inability of models to
capture lower stratospheric trends, i.e the biases are introduced somewhere else in the system
unrelated to chemistry and stratospheric dynamics? If one ensemble member can reproduce
historic trends, is it then in the realm of possibilities in the modeling world to reproduce observed
trends, so to speak? If yes, can you still come to such strong conclusions concerning the
models' skill to reproduce past trends?

The reviewer raises important questions here. We investigated the N3.4 metric estimated from
the lowest level air temperature in the models and compare these with observations. This is
addressed earlier in the response to reviewers, above. We added a new figure (Fig. S1) and
some additional discussion.

Other comments:

- l.9: 'an increase'

Done.

- l.96-109: see above. Do we expect the MMM to be able to reproduce such a short period of
time on average, or are we looking for individual ensemble members for this?

See response above.

- l.138: typo?

Could not identify typo.

- l.179: any particular reason why you used those 12?

This was elucidated in the model discussion in section 2.2. Fundamentally this came down to what was available on the BADC server. Some models did not have all variables available, or e.g. for GEOSCCM data were only available from 2002. UMUKCA-METO has subsequently become (re-)available and we have added this in.

- Figure 1: why are units/scalings for (c) and (d) not consistent?

The aim is to compare the shape and sign of the different curves. But the reviewer is right that they differ between observations and models. The issue is the second derivative in modelled ozone is *five times larger* than in the observations, but approximately *two times smaller* in temperature and water vapour. Therefore, it was rescaled to fit.

- l.235-255: I am still not convinced that the MMM should agree with the single member observations. Do you have any theoretical justification for that?

Ensemble means should capture the model-dependent forced response, while the MMM gives the mean model forced response. So, there is no reason why a single ensemble member, especially on a relatively short timescale, should necessarily agree. Our point here is to emphasize that the MMM, used for recovery date estimates in international reports (e.g. WMO ozone assessment) use the MMM, but that the MMM differs from the observations. It is also explicitly why we include and show the individual models where feasible given the need to make a coherent narrative. We have added points to the manuscript considering this point, some of which have been given in response to the first reviewer.

- l.283: typo

Fixed.

---

## Author Response (AR2)

**In response to the editors comments**

We thank the editor for her thorough consideration of our manuscript and revisions, and for her additional suggestions. We have implemented all of these, and in certain cases we respond as below (black: editor's comments; blue our comments).

Please check the references for the figures. Since you go quite often back and forth and point on specific panels it is a bit difficult to keep track of all of them. At some occasion I had the feeling that the referencing was not entirely correct (see my comments below).

Agreed – I also found some additional mistakes and have corrected these. I have also reordered the SM figures to reflect the order they are mentioned in the main article.

There are many acronyms used throughout the text and I was wondering if it could be worth to provide a list of acronyms as appendix in the manuscript or in the supplement to make it a bit easier for the reader.

Agreed: I have added a comment to the list being in the supplementary materials in the final paragraph of the introduction, and added a list to the beginning of the Supplementary materials.

P4, L79: "….and 2018…" does the year 2018 refer to another Ball et al. paper? If yes, the parentheses should be moved.

No, but I have modified the text to flow a bit better!

P7, L188: Instead of "remain appropriate" I would express that a bit differently. My suggestions would be " best option" , "best choice" or "best suitable data set".

I actually struggled over these two words before! I have gone with "best option"; thanks for the suggestion.

P8, L248: The acronym DLM has been introduced, but I think DLMMC not.

There is no specific acronym, so I have written "the code called 'DLMMC'"

P8, L249: I am not sure if the acronym MLR has been introduced.

It had, in the previous section.

P10, L289: Here, I was not sure if you really mean Fig. S3-S5. I would have said only Fig S4 since there you show different latitude bands.

This is meant to provide a sensitivity comparison of the 50-50 and 60-60 bands, so I think there is indeed an error here, and S4 should be dropped, which I do.

P15, L354: Figs. 4a and c -> Figs. 4a-c?:

4b is not included in the reference (as it discusses mixing), so we left this as is.

P16, L411: The differences are hard to see. Quickly looking at the figure and comparing them one comes to the conclusion they almost look the same. Of course, if one looks directly at the observations in the plot, it gets quickly obvious. Here, it could be helpful to guide the a bit more since the observations go a bit under in the crowd of the large number of models.

This is fair; I have added "(compare orange and black on the right side of the panel)" to the end of the sentence.

P19, L456: Really both hemispheres? I see this behavior rather for the NH than SH.

I am not sure I understand this: the zero line goes through the shading in observations so there is little confidence in a change. Is that what you refer to?